# A Robust $\widetilde{\mathcal{O}}(1/\sqrt{T})$ Rate for Unprojected TD Learning with Linear Function Approximation

## Abstract

We investigate the finite-time convergence properties of Temporal Difference (TD) learning with linear function approximation, a cornerstone of reinforcement learning. We are interested in the so-called "robust" setting, where the convergence guarantee does not depend on the potential function's minimal curvature. While prior work has established convergence guarantees in this setting, these results typically rely on the artificial assumption that each iterate is projected onto a bounded set. Removing such a condition was left as an open problem by Bhandari et al. (COLT'18), hypothesizing the need for additional "regularity conditions". In this paper, we show that the simple unprojected TD(0) converges with a rate of $\widetilde{\mathcal{O}}\left(\frac{\|\boldsymbol{\theta}^*\|_2^2}{\sqrt{T}}\right)$ in expectation, even in the presence of Markovian noise. We do not require an additional regularity condition, but only a minor polylog correction to the learning rate. Our analysis reveals a novel self-bounding property of the TD updates and exploits it to guarantee bounded iterates.

## 1 Introduction

Temporal Difference (TD) learning (Sutton, 1988) is a cornerstone of modern reinforcement learning. It provides a model-free approach to policy evaluation, estimating the value function of a given policy within a Markov Decision Process (MDP). The versatility of TD methods has led to applications in diverse domains, including games (Silver et al., 2016), robotics (Gu et al., 2017), and autonomous systems (Chen et al., 2015). At its core, TD learning iteratively updates value function estimates based on the difference between predictions at successive time steps.

Despite its conceptual simplicity and widespread use, the theoretical analysis of TD learning, particularly with linear function approximation in large state spaces, presents considerable challenges. Early seminal work by Tsitsiklis & Van Roy (1996) established asymptotic convergence conditions by framing TD as a stochastic approximation algorithm. More recently, understanding the non-asymptotic behavior and finite-time performance of TD has become an active area of research. Challenges primarily arise from the correlated nature of samples generated by the underlying Markov chain, which can introduce bias and dependencies into the learning updates.

Several studies have provided finite-time analyses under various assumptions on the potential function, the projection step, and the stepsize. In particular, two *complementary* kinds of analyses are known, giving rise to a "robust" convergence rate of $\widetilde{\mathcal{O}}(1/\sqrt{T})$[1] (e.g., Bhandari et al., 2018; Liu & Olshevsky, 2021; Sun et al., 2021) or to a "fast" rate of $\widetilde{\mathcal{O}}(1/T)$ (e.g., Bhandari et al., 2018; Srikant & Ying, 2019; Patil et al., 2023; Samsonov et al., 2024; Mitra, 2024; Li et al., 2025). These two rates are complementary because the hidden constant in the fast rate depends on the inverse square of the curvature of the potential function which, while always present, can be arbitrarily small. Instead, the $\widetilde{\mathcal{O}}(1/\sqrt{T})$ robust rate is independent of the curvature. Hence, in non-asymptotic regimes, the fast rate can be arbitrarily worse than the robust one.[2] This mirrors

---

[1] $\widetilde{\mathcal{O}}$ hides polylogarithmic terms and may also hide dependencies on the mixing time.
[2] See Appendix A for an in-depth discussion of the literature on this point.

Table 1: Summary of algorithmic inputs and rates for TD(0) with linear function approximation in the literature. The quantity $\omega$ is the minimum eigenvalue of $\mathbf{\Phi}^\top \boldsymbol{D} \mathbf{\Phi}$. All quantities are defined in Section 3 and their inputs are discussed in Appendix B.

| Rate | Paper | Inputs | Without Projection | Bound independent of $\omega$ |
|---|---|---|---|---|
| $\widetilde{\mathcal{O}}\left(\frac{1}{T}\right)$ | Bhandari et al. (2018) | $\omega, \phi_\infty$ | ✗ | ✗ |
| | Srikant & Ying (2019) | $\omega, \alpha, \phi_\infty, \|\boldsymbol{\theta}^*\|_2$ | ✓ | ✗ |
| | Patil et al. (2023) | $\alpha, \phi_\infty$ | ✓ | ✗ |
| | Samsonov et al. (2024) | $\alpha, \phi_\infty$ | ✓ | ✗ |
| | Mitra (2024) | $\omega, \alpha, \phi_\infty$ | ✓ | ✗ |
| | Li et al. (2025) | $\omega, \phi_\infty$ | ✓ | ✗ |
| $\widetilde{\mathcal{O}}\left(\frac{1}{\sqrt{T}}\right)$ | Bhandari et al. (2018) | $\phi_\infty$ | ✗ | ✓ |
| | Liu & Olshevsky (2021) | $\phi_\infty$ | ✗ | ✓ |
| | Sun et al. (2021) | $\phi_\infty$ | ✗ | ✗ |
| | **This paper**, Theorem 4.2 | $\phi_\infty$ | ✓ | ✓ |

what happens in the stochastic approximation setting, and it is well-known that in practice the robust rate can be preferable, as motivated in Nemirovski et al. (2009).

In this work, we focus on the need for a projection step to achieve robust rates. In fact, for fast rates, the assumption of minimal curvature leads to a contraction that simplifies the analysis, eliminating the need for a projection. However, there are no known results on unprojected TD achieving the robust $\widetilde{\mathcal{O}}(1/\sqrt{T})$ rate, whereas in practice such a projection is never used. It is worth stressing that while such a projection step is not used in practice, but only to simplify the theoretical analysis.[3] Indeed, Bhandari et al. (Section 11 2018) explicitly posed the removal of the projection step as an open problem, in both the fast and robust regime, hypothesizing that it would be possible "under additional regularity conditions." While removing the projection for the fast case was solved after one year by Srikant & Ying (2019), the unprojected robust case remained unresolved.

**Contributions.** In this paper, we solve one of the open problems posed by Bhandari et al. (2018): For the first time, we provide a finite-time analysis of TD(0) with linear function approximation under Markovian observations *without requiring iterate projection, while achieving a robust rate.* Moreover, we do not require any additional regularity conditions. Instead, we show that changing the learning rate from $\frac{1}{\sqrt{T}}$ or $\frac{1}{\sqrt{t}}$ to $\frac{1}{\sqrt{t}\log t \log T}$ is sufficient to guarantee a self-bounding property of TD: The iterates are constrained, in expectation, to a bounded domain around the optimal solution. Our analysis differs fundamentally from those that aimed to prove the update is a noisy contraction, and it might be of independent interest. We also show a convergence rate $\widetilde{\mathcal{O}}(\frac{\|\boldsymbol{\theta}^*\|_2^2}{\sqrt{T}})$ for the potential that guides the convergence of the TD algorithm, as defined in Liu & Olshevsky (2021). Table 1 summarizes[4] and compares our results with existing finite-time analyses of TD with linear function approximation.

## 2  Related Work

The initial theoretical understanding of how TD learning with linear function approximation converges over time was established by Tsitsiklis & Van Roy (1996), who framed TD methods as stochastic approximation algorithms (Kushner, 2010). That work did not derive finite-time convergence rates. Subsequent research (Korda & La, 2015; Lakshminarayanan & Szepesvári, 2018; Dalal et al., 2018) did provide such rates, but a significant limitation was the assumption that data are drawn independently from the stationary distribution. In practice, data are typically collected sequentially along a single trajectory of the Markov chain, introducing

---

[3]Bhandari et al. (2018) says "at this stage, we view this [projection] mainly as a tool that enables clean finite time analysis, rather than a practical algorithmic proposal."

[4]See Appendix B for a precise discussion of the inputs for each algorithm.

temporal correlations between samples. These correlations make it challenging to analyze even the basic TD(0) method.

Bhandari et al. (2018) provided the first finite-time analysis of TD learning under more realistic Markovian data, drawing parallels to stochastic gradient descent. However, their analysis, as well as that of Liu & Olshevsky (2021), requires a projection step to control the magnitude of iterates/updates. Sun et al. (2021) examined Adam-inspired (Kingma, 2014) adaptive TD variants, but they require a projection as well. Here, we remove the need to project, while obtaining the robust rate of $\widetilde{\mathcal{O}}(1/\sqrt{T})$ obtained by Bhandari et al. (2018).

Another line of work leverages the curvature of the potential function. This allowed Srikant & Ying (2019) to be the first to provide finite-time error bounds for TD learning with linear function approximation under Markovian sampling without a projection step, employing a control-theoretic approach based on Lyapunov theory. While elegant, the analysis in Srikant & Ying (2019) relies on stepsizes that depend on the strong-convexity (curvature) parameter of the potential function. Since this parameter is typically unknown, their result only implies the existence of a good but unknown learning rate. Subsequently, Patil et al. (2023) removed the dependence of stepsizes on this strong-convexity parameter, yielding a more practical algorithm, but at the price of requiring a data-dropping variant of TD. Later, Samsonov et al. (2024) improved the analysis of Patil et al. (2023) to obtain high-probability bounds.

In parallel, Mitra (2024) provided a simpler analysis using an inductive two-step argument, while Li et al. (2025) utilized an exponentially decaying stepsize to remove the data-dropping steps. Moreover, Sun et al. (2022) extended the fast analysis to neural networks in the NTK regime. However, in all these results, the non-asymptotic convergence rate can become arbitrarily slow due to ill-conditioned linear mappings. We discuss this caveat in more detail in Section 4.1.

Our proof method is fundamentally different from the above ones, which removed the projection by proving a contraction. Instead, we show that the iterates are bounded for reasons analogous to what happens in Stochastic Gradient Descent (SGD). In fact, SGD can have bounded iterates even for non-strongly convex objectives, as shown, for example, by Xiao (2010); Orabona & Pál (2021); Telgarsky (2022); Ivgi et al. (2023) under various update schemes and assumptions on the potential and stepsizes.

Another minor difference with prior work is our choice of the potential function: We study the potential function proposed in Liu & Olshevsky (2021), which improves earlier formulations by adding a term proportional to the discount factor $\gamma$.

## 3 Notation and Assumptions

We briefly review the required background on Markov Decision Processes (MDPs) and TD learning with linear function approximation. For a comprehensive treatment, we refer the reader to Sutton & Barto (1998) and Mannor et al. (2022).

In the following, vectors are denoted in bold, and all norms are $\ell_2$ (i.e., Euclidean) norms unless stated otherwise.

### 3.1 Discounted Markov Decision Processes

We define a *discounted-reward MDP* as a tuple $(\mathcal{S}, \mathcal{A}, P, r, \gamma)$ ... transition kernel $P$, where $P(s' \mid s, a)$ denotes the probability of transitioning from $s$ to $s'$ given action $a$. The reward $r(s, s')$ for each transition is bounded by $r_\infty$. Given a trajectory starting at $s_0$, we denote the state at time $t$ by $s_t$, with the reward after transitioning defined as $r_t := r(s_t, s_{t+1})$.

**Induced Markov chain.** A stationary policy $\mu : \mathcal{S} \to \Delta^{|\mathcal{A}|-1}$ induces a Markov chain with transition probabilities

$$P^\mu(s' \mid s) := \sum_{a \in \mathcal{A}} \mu(a \mid s) P(s' \mid s, a), \quad \forall s, s' \in \mathcal{S}.$$

We let $\boldsymbol{P}^\mu \in \mathbb{R}^{n \times n}$ denote the corresponding transition matrix, where the entry at row $s$ and column $s'$ is $[\boldsymbol{P}^\mu]_{s,s'} = P^\mu(s' \mid s)$. Throughout, we denote the expected single-step reward at state $s$ by

$$\bar{r}(s) := \sum_{s'} P^\mu(s' \mid s)\, r(s, s').$$

In this paper, we focus on the task of *policy evaluation*, where the goal is to compute the value function defined as the expected discounted sum of rewards.

**Value functions and Bellman operators.** The value function $\boldsymbol{V}^\mu \in \mathbb{R}^n$ associated with policy $\mu$ is defined component-wise as $\boldsymbol{V}^\mu(s) = \mathbb{E}[\sum_{t=0}^\infty \gamma^t\, r_t \mid s_0 = s]$, where the expectation is taken over the trajectory generated by $\boldsymbol{P}^\mu$ starting from $s_0 = s$. The Bellman operator $\mathcal{T}^\mu : \mathbb{R}^n \to \mathbb{R}^n$ is defined as

$$(\mathcal{T}^\mu \boldsymbol{V})(s) := \bar{r}(s) + \gamma \sum_{s' \in \mathcal{S}} P^\mu(s' \mid s)\, \boldsymbol{V}(s'), \quad \forall s \in \mathcal{S}.$$

The operator $\mathcal{T}^\mu$ is a $\gamma$-contraction in the $\ell_\infty$-norm; hence $\boldsymbol{V}^\mu$ is its unique fixed point.

To study the finite-time behavior of the Markov chain, we impose the following standard ergodic condition.

**Assumption 3.1.** The Markov chain induced by policy $\mu$ is irreducible and aperiodic.

Under Assumption 3.1, the Markov chain admits a unique stationary distribution $\pi \in \Delta^{n-1}$ and its vector form $\boldsymbol{\pi}$ satisfies $\boldsymbol{\pi}^\top \boldsymbol{P}^\mu = \boldsymbol{\pi}^\top$ and mixes geometrically:

**Theorem 3.2** (Levin & Peres 2017, Theorem 4.9). *There exist constants $1 < C \le 2$ and $\alpha \in [1/2, 1)$ such that*

$$\max_{s \in \mathcal{S}} \left\| (P^\mu)^t(\cdot \mid s) - \pi \right\|_{\mathrm{TV}} \le C\,\alpha^t, \quad \forall t \ge 0,$$

*where $\| \cdot \|_{\mathrm{TV}}$ denotes the total variation distance and $(P^\mu)^t(\cdot \mid s)$ is the state distribution after $t$ steps starting from state $s$.*

Based on this, we define the *mixing time* for a tolerance $\epsilon$ as $\tau(\epsilon) := \min\{t \in \mathbb{N} \mid C\alpha^t \le \epsilon\}$.

### 3.2 TD(0) with Linear Function Approximation

We consider the approximation of $\boldsymbol{V}^\mu$ with linear mappings and estimate the weights $\boldsymbol{\theta} \in \mathbb{R}^d$ via TD learning.

**Linear architecture.** Let $\phi_i : \mathcal{S} \to \mathbb{R}$ for $i \in \{1, \ldots, d\}$ be the feature mappings. For each $s \in \mathcal{S}$, define the feature vector $\boldsymbol{\phi}(s) := [\phi_1(s), \ldots, \phi_d(s)]^\top \in \mathbb{R}^d$. We define the feature matrix $\boldsymbol{\Phi} \in \mathbb{R}^{n \times d}$ such that the row corresponding to state $s$ is $\boldsymbol{\phi}(s)^\top$. The value function is approximated as $V_{\boldsymbol{\theta}}(s) := \boldsymbol{\theta}^\top \boldsymbol{\phi}(s)$, or in vector form $\boldsymbol{V_\theta} = \boldsymbol{\Phi}\boldsymbol{\theta}$.

We recall the following standard assumption on the features.

**Assumption 3.3.** The feature matrix $\boldsymbol{\Phi}$ has full column rank $d$, and $\|\boldsymbol{\phi}(s)\| \le \phi_\infty$ for[5] all $s \in \mathcal{S}$.

**TD error and update.** For $z = (s, s') \in \mathcal{S} \times \mathcal{S}$, define the TD update map

$$\boldsymbol{g}(\boldsymbol{\theta}, z) := \left( r(s, s') + \gamma \boldsymbol{\phi}(s')^\top \boldsymbol{\theta} - \boldsymbol{\phi}(s)^\top \boldsymbol{\theta} \right) \boldsymbol{\phi}(s). \tag{1}$$

For $Z_t := (s_t, s_{t+1})$, we write

$$\boldsymbol{g}_t := \boldsymbol{g}(\boldsymbol{\theta}_t, Z_t).$$

Given the weight $\boldsymbol{\theta}_t$ and a trajectory sample $(s_t, s_{t+1}, r_t)$, the TD error is defined as $\delta_t := r_t + \gamma V_{\boldsymbol{\theta}_t}(s_{t+1}) - V_{\boldsymbol{\theta}_t}(s_t)$. The TD(0) updates as

$$\boldsymbol{\theta}_{t+1} = \boldsymbol{\theta}_t + \eta_t\, \delta_t\, \nabla_{\boldsymbol{\theta}} V_{\boldsymbol{\theta}_t}(s_t) = \boldsymbol{\theta}_t + \eta_t \left( r_t + \gamma \boldsymbol{\theta}_t^\top \boldsymbol{\phi}(s_{t+1}) - \boldsymbol{\theta}_t^\top \boldsymbol{\phi}(s_t) \right) \boldsymbol{\phi}(s_t) := \boldsymbol{\theta}_t + \eta_t \boldsymbol{g}_t,$$

where $\eta_t > 0$ is the stepsize.

---

[5]The knowledge of $\phi_\infty$ can be removed by using, for example, a feature normalization scheme.

---

**Algorithm 1** Unprojected TD(0) with linear function approximation

---
1: **Input:** iteration budget $T$, $\phi_\infty$, $c > 15 + 18\sqrt{2}$, $s_0$
2: $\boldsymbol{\theta}_0 = \mathbf{0}$
3: **for** $t = 0, \ldots, T-1$ **do**
4:     Receive trajectory sample $(s_t, s_{t+1}, r_t)$
5:     $\boldsymbol{g}_t = \left(r_t + \gamma\boldsymbol{\phi}(s_{t+1})^\top\boldsymbol{\theta}_t - \boldsymbol{\phi}(s_t)^\top\boldsymbol{\theta}_t\right)\boldsymbol{\phi}(s_t)$
6:     Set the stepsize

$$\eta_t = \frac{1}{c\,\phi_\infty^2\,\log(T)\,\log(t+3)\,\sqrt{t+1}}$$

7:     $\boldsymbol{\theta}_{t+1} = \boldsymbol{\theta}_t + \eta_t\boldsymbol{g}_t$
8: **end for**
9: **Output:** $\bar{\boldsymbol{\theta}}_T := \left(\sum_{k=0}^{T-1}\eta_k\right)^{-1}\sum_{k=0}^{T-1}\eta_k\boldsymbol{\theta}_k$

---

**Projected Bellman equation.** Under Assumptions 3.1–3.3 and suitable $\eta_t$, TD(0) converges to $\boldsymbol{\theta}^*$ asymptotically and $\boldsymbol{\theta}^*$ is characterized as the unique solution to the projected Bellman equation (Tsitsiklis & Van Roy, 1996) $\boldsymbol{\Phi}\boldsymbol{\theta}^* = \Pi_{\boldsymbol{D}}\mathcal{T}^\mu(\boldsymbol{\Phi}\boldsymbol{\theta}^*)$, where $\Pi_{\boldsymbol{D}}$ is the orthogonal projection operator onto the subspace $\{\boldsymbol{\Phi}\boldsymbol{x} \mid \boldsymbol{x} \in \mathbb{R}^d\}$ with respect to the $\boldsymbol{D}$-norm defined below.

**D-norms.** Let $\boldsymbol{D} := \mathrm{diag}(\boldsymbol{\pi})$. Since $\boldsymbol{D} \succ \mathbf{0}$, for $\boldsymbol{x}, \boldsymbol{y} \in \mathbb{R}^n$, we can define the inner product $\langle\boldsymbol{x}, \boldsymbol{y}\rangle_{\boldsymbol{D}} := \boldsymbol{x}^\top\boldsymbol{D}\boldsymbol{y}$ and the norm $\|\boldsymbol{x}\|_{\boldsymbol{D}} := \sqrt{\langle\boldsymbol{x}, \boldsymbol{x}\rangle_{\boldsymbol{D}}}$.

**Dirichlet-seminorms.** The *Dirichlet seminorm* (Diaconis & Saloff-Coste, 1996; Ollivier, 2018; Liu & Olshevsky, 2021) is defined as

$$\|\boldsymbol{V}\|_{\mathrm{Dir}}^2 := \frac{1}{2}\sum_{s,s'\in\mathcal{S}}\pi(s)\,P^\mu(s' \mid s)\big(V(s') - V(s)\big)^2. \tag{2}$$

Since $\boldsymbol{\Phi}$ is full column rank by Assumption 3.3, the matrix $\boldsymbol{\Phi}^\top\boldsymbol{D}\boldsymbol{\Phi}$ is positive definite. Thus, the minimum eigenvalue $\omega := \lambda_{\min}(\boldsymbol{\Phi}^\top\boldsymbol{D}\boldsymbol{\Phi})$ is strictly positive. This quantity $\omega$ plays the role of a strong-convexity (curvature) parameter in fast-rate analyses.

**Stationary update.** The asymptotic behavior of TD(0) is closely tied to the stationary update field

$$\bar{\boldsymbol{g}}(\boldsymbol{\theta}) := \mathbb{E}\big[\big(r(s, s') + \gamma\boldsymbol{\phi}(s')^\top\boldsymbol{\theta} - \boldsymbol{\phi}(s)^\top\boldsymbol{\theta}\big)\boldsymbol{\phi}(s)\big].$$

Here, the expectation is taken over stationary transitions where $s \sim \pi$ and $s' \sim P^\mu(\cdot \mid s)$. In particular, $\bar{\boldsymbol{g}}(\boldsymbol{\theta}^*) = \mathbf{0}$.

# 4 Unprojected Temporal Difference Learning

In this section, we present our main result: We analyze TD(0) without any projection, as shown in Algorithm 1, and present a robust convergence result for it.

First, we explain what exactly our convergence is. In prior work (e.g., Bhandari et al., 2018), the potential function for the convergence analysis was

$$\big\|\boldsymbol{V}_{\bar{\boldsymbol{\theta}}_T} - \boldsymbol{V}_{\boldsymbol{\theta}^*}\big\|_{\boldsymbol{D}}^2. \tag{3}$$

Note that when the discount factor $\gamma \to 1$, their rate $\widetilde{\mathcal{O}}\left(\frac{\|\boldsymbol{\theta}^*\|^2}{(1-\gamma)\sqrt{T}}\right)$ loses its utility in characterizing the error of estimates $\bar{\boldsymbol{\theta}}_T$ to $\boldsymbol{\theta}^*$ in $T$. For this reason, we focus on the better potential function proposed by Liu & Olshevsky (2021):

$$f(\boldsymbol{\theta}) := (1-\gamma)\|\boldsymbol{V}_{\boldsymbol{\theta}} - \boldsymbol{V}_{\boldsymbol{\theta}^*}\|_{\boldsymbol{D}}^2 + \gamma\|\boldsymbol{V}_{\boldsymbol{\theta}} - \boldsymbol{V}_{\boldsymbol{\theta}^*}\|_{\mathrm{Dir}}^2.$$

Clearly, the point $\boldsymbol{\theta}^*$ minimizes $f(\boldsymbol{\theta})$. Moreover, any result obtained using this potential with $(1-\gamma)^{-1}$ scaling implies those obtained using (3) since $\|\boldsymbol{V_\theta} - \boldsymbol{V_{\theta^*}}\|_{\text{Dir}}^2$ is non-negative. Finally, this potential provides convergence results even when the discount factor $\gamma \to 1$; see the discussion in Liu & Olshevsky (2021).

From a technical point of view, the advantage of this potential is that the stationary update $\bar{\boldsymbol{g}}(\boldsymbol{\theta})$ satisfies the equality in the following theorem. In a sense, the results in Liu & Olshevsky (2021) indicate that this is the "correct" potential function for TD(0).

**Lemma 4.1.** *(Liu & Olshevsky, 2021, Theorem 1) For any $\boldsymbol{\theta} \in \mathbb{R}^d$, we have*

$$\langle -\bar{\boldsymbol{g}}(\boldsymbol{\theta}), \boldsymbol{\theta} - \boldsymbol{\theta}^* \rangle = f(\boldsymbol{\theta}) - f(\boldsymbol{\theta}^*).$$

Using the above potential function, the following theorem shows that the TD(0) algorithm can converge with a rate of $\widetilde{\mathcal{O}}(\|\boldsymbol{\theta}^*\|^2/\sqrt{T})$ without projections.

**Theorem 4.2.** *Consider the weighted average iterate $\bar{\boldsymbol{\theta}}_T$ generated by Algorithm 1. Suppose the stepsize parameter $c$ satisfies $c > c_0 := 15 + 18\sqrt{2}$, and the number of iterations $T$ satisfies $\log T \geq \frac{2}{\log^3(1/\alpha)}$. Then, we have, for any $t \leq T$,*

$$\mathbb{E}\left[\|\boldsymbol{\theta}_t\|^2\right] \leq \rho_c^2 \max\left\{\frac{r_\infty^2}{\phi_\infty^2}, \|\boldsymbol{\theta}^*\|^2\right\},$$

*where $\rho_c \to 2$ as $c \to \infty$, and $\rho_c = \mathcal{O}\left(\frac{1}{c-c_0}\right)$ as $c \downarrow c_0$.*

**Where the threshold on $c$ come from.** The threshold $c > c_0 = 15 + 18\sqrt{2}$ should be viewed as a conservative sufficient condition for the proof, not as a practical tuning rule. This constant comes from the self-bounding induction used to prove bounded iterates for $t \leq T$. More specifically, it accumulates from Theorem 3.2, the use of the Integral test, repeated uses of the triangle inequality, Cauchy–Schwarz, and AM–GM type inequalities. Thus, $c$ is chosen large enough so that all these proof-level constants can be absorbed and the induction closes. The radius multiplier $\rho_c$ is determined only by $c$. In particular, $\rho_c$ does not depend on the mixing parameter $\alpha$ or the discount factor $\gamma$.

**Corollary 4.3.** *Under the same assumptions as Theorem 4.2, we have*

$$f(\bar{\boldsymbol{\theta}}_T) - f(\boldsymbol{\theta}^*) = \widetilde{\mathcal{O}}\left(\frac{c\rho_c^2 \max\left\{r_\infty^2, \phi_\infty^2 \|\boldsymbol{\theta}^*\|^2\right\}}{\sqrt{T}}\right).$$

*Here, $\widetilde{\mathcal{O}}$ hides only logarithmic factors in $T$; in particular, it hides no dependence on the mixing constant $\alpha$ or the discount factor $\gamma$.*

For lack of space, we only give the proof sketch of Theorem 4.2 in Section 5, where we explain the main steps of the proof and contrast it with previous proofs. The full proof of Theorem 4.2 is given in Appendix G.1, while the proof of Corollary 4.3 is given at the end of Section 5. Theorem G.1 will also detail how the stepsize parameter $c$ determines the radius multiplier $\rho_c$. In short, the number $c$ must exceed a threshold to ensure bounded iterates, and increasing $c$ decreases $\rho_c$.

It is worth stressing that knowing $T$ in advance is not a limitation, because one can use a standard doubling trick (see Appendix J).

**Dependency on $\alpha$ for $T$?** The condition on $T$ in Theorem 4.2 is used to absorb the mixing-related terms arising from Markovian sampling, thereby ensuring that the induction closes. Since $\alpha$ is typically unknown, one may question the practicality of this condition. First, we emphasize that this is a limitation of all analyses in this line of work, although it is sometimes hidden. Indeed, all prior robust bounds yield a rate worse than $\widetilde{\mathcal{O}}(1/\sqrt{T})$ unless $T$ is sufficiently large as a function of $\alpha$. This point appears to be poorly discussed in the literature, so we devote Appendix I to proving it. It is possible to state our result in an anytime form, but at the very least, the stepsize would then depend on unknown $\alpha$; we therefore prefer a more executable presentation of the assumptions and results.

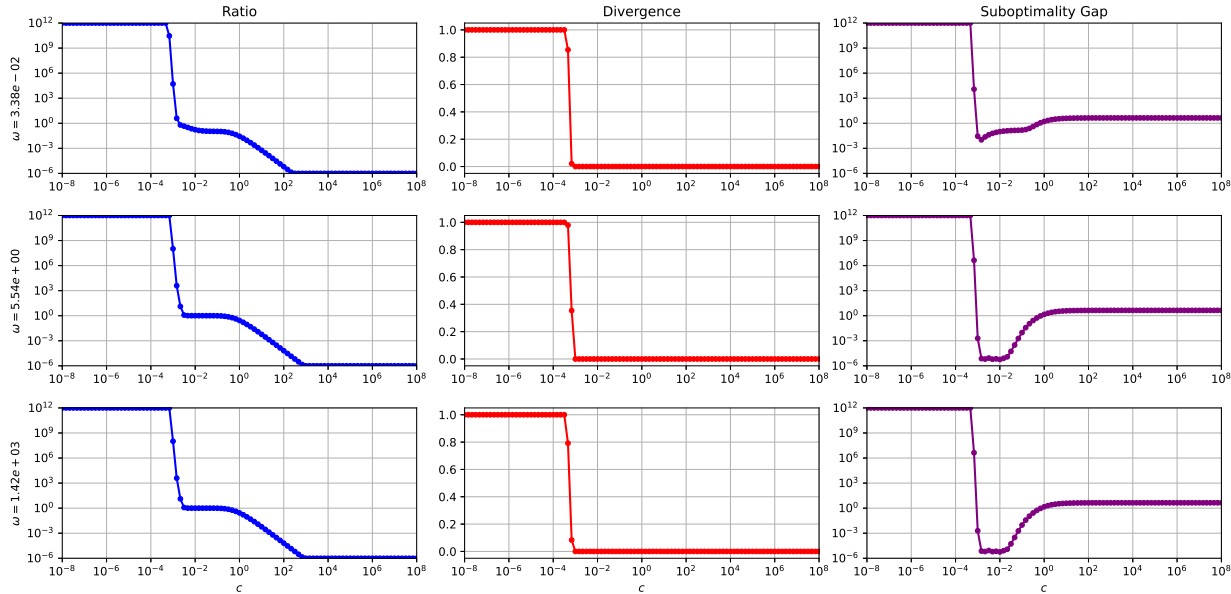

Figure 1: Instability of TD learning. Columns: boundedness ratio, divergence rate, suboptimality gap vs. stepsize scale (c). Rows: different feature scalings (changing the spectrum of $\boldsymbol{\Phi}^\top \boldsymbol{D}\boldsymbol{\Phi}$).

Notice that even for fast rates, one either needs a sufficiently small stepsize that depends on $\omega$ (an unknown quantity), or must use $\omega$-agnostic algorithms (e.g., Patil et al., 2023; Samsonov et al., 2024), which rely on data-dropping steps that still require knowledge of $\alpha$; see Table 1.

**Is the threshold on the stepsizes real?**   Theorem 4.2 gives a sufficient condition on $c$ to have bounded iterates and convergence. First of all, we would like to stress that we did not try to optimize the numerical value of the threshold on $c$, for the simple reason that any similar analysis with such weak assumptions cannot be predictive of reality.

Instead, the more interesting question is to check if the condition is necessary too. That is, does TD(0) without projection have bounded iterates in finite time with arbitrary stepsizes satisfying the condition in Tsitsiklis & Van Roy (1996)? To test this effect, we conducted an experiment in which we ran TD(0) on a synthetic problem (details in Appendix H). In Figure 1, first column, we show the expected boundedness ratio, defined as $\frac{\max_{t \leq T} \mathbb{E}\left[\|\boldsymbol{\theta}_t\|^2\right]}{\|\boldsymbol{\theta}^*\|^2}$, which is large if the iterates blow up. The second column shows the divergence rate, that is, the fraction of runs with $\|\boldsymbol{\theta}_t\|^2 > 10^{12}$, while the third column shows the suboptimality gap. Overall, these results suggest that the threshold on the stepsizes captures a genuine finite-time stability effect. In fact, from both the expected boundedness ratio and the divergence rate, it is clear that if $c$ is too small, the iterates of the algorithm are not controlled in finite time. Moreover, the explosion in both of these measures is qualitatively consistent with the theoretical behavior encoded by our function $\rho_c$ in Theorem 4.2.

The rows correspond to different spectral characteristics of $\boldsymbol{\Phi}^\top \boldsymbol{D}\boldsymbol{\Phi}$. Hence, changing the spectral characteristics does not change much the iterates, as our theory suggests. However, due to the complementarity of the robust and fast rates, it does influence the suboptimality gap, suggesting that a robust rate of $\widetilde{\mathcal{O}}(1/\sqrt{T})$ is pessimistic when the strong convexity is large.

In Appendix H, we also report experiments with a fixed stepsize that show the same behaviors.

## 4.1   Detailed Comparison with Previous Results

Here, we highlight a few technical differences between our analysis and existing finite-time results for TD(0) with linear function approximation.

**Comparison with robust $\widetilde{\mathcal{O}}(1/\sqrt{T})$ rates with projections.** Bhandari et al. (2018, Theorem 3.(a)) proves that *projected* TD with stepsize $\eta_t = \frac{1}{\sqrt{T}}$ satisfies

$$\mathbb{E}\big[\|\boldsymbol{V}_{\bar{\boldsymbol{\theta}}_T} - \boldsymbol{V}_{\boldsymbol{\theta}^*}\|_{\boldsymbol{D}}^2\big] = \widetilde{\mathcal{O}}\bigg(\frac{R^2}{(1-\gamma)\sqrt{T}}\bigg),$$

where $R$ is the projection radius and requires $R \geq \|\boldsymbol{\theta}^*\|$. Our bound implies the same type of $D$-norm guarantee, since $\|\cdot\|_{\mathrm{Dir}}^2 \geq 0$. Also, their choice of the learning rate and ours have the same dependency on $T$, up to polylogarithmic terms, that is, we do not gain more stability by using a much smaller learning rate, but rather with a refined analysis. Moreover, our bound depends explicitly on $\|\boldsymbol{\theta}^*\|$ rather than on an a priori radius $R$.

We stress once again that the use of a projection step in Bhandari et al. (2018) is only for the purpose of analysis, but not a real practical possibility. In fact, while one can obtain a (potentially very loose) upper bound on $\|\boldsymbol{\theta}^*\|$ in terms of $\omega = \lambda_{\min}(\boldsymbol{\Phi}^\top \boldsymbol{D}\boldsymbol{\Phi})$ (Bhandari et al., 2018, Lemma 1), this approach is impractical from an algorithm design perspective. In fact, the goal of TD learning is precisely to avoid the computational complexity that scales with the number of states $n$, which is highly non-trivial when estimating $\omega$ involves computing the stationary matrix $\boldsymbol{D}$.

**Comparison with fast $\widetilde{\mathcal{O}}(1/T)$ rates.** It is instructive to compare our results with the known fast rates, to underline their complementarity. In the fast regime (Srikant & Ying, 2019; Patil et al., 2023; Mitra, 2024; Samsonov et al., 2024; Li et al., 2025), contraction-based arguments yield bounds of the form

$$\mathbb{E}\big[\|\boldsymbol{V}_{\bar{\boldsymbol{\theta}}_T} - \boldsymbol{V}_{\boldsymbol{\theta}^*}\|_{\boldsymbol{D}}^2\big] = \widetilde{\mathcal{O}}\bigg(\frac{\|\boldsymbol{\theta}^*\|^2}{(1-\gamma)^2\,\omega^2\,T}\bigg) .$$

Some algorithms choose stepsizes without prior knowledge of $\omega$ (Patil et al., 2023; Samsonov et al., 2024), matching the same information passed to the analyses achieving a robust rate. Nevertheless, in non-asymptotic regimes, the fast guarantee can be much worse than an $\omega$-independent $\widetilde{\mathcal{O}}(1/\sqrt{T})$ guarantee, because $\omega$ can be arbitrarily small. Indeed, ignoring logarithmic factors, the fast bound only improves over $\widetilde{\mathcal{O}}(\|\boldsymbol{\theta}^*\|^2/((1-\gamma)\sqrt{T}))$ when $T \gtrsim 1/((1-\gamma)^2\omega^4)$. We now show that one can construct problems where $\omega$ is arbitrarily small.

Consider a two-state Markov chain with states $s_1$ and $s_2$, and transition matrix

$$\boldsymbol{P} = \begin{bmatrix} \frac{1-\alpha}{2} & \frac{1+\alpha}{2} \\ \frac{1+\alpha}{2} & \frac{1-\alpha}{2} \end{bmatrix}, \qquad \frac{1}{2} < \alpha < 1 .$$

Then, this chain is irreducible and aperiodic, and its stationary distribution is uniform: $\boldsymbol{D} = \frac{1}{2}\mathbf{I}$. Now consider the feature matrix

$$\boldsymbol{\Phi} = \begin{bmatrix} \epsilon & 1 \\ -\epsilon & 1 \end{bmatrix}, \qquad 0 < \epsilon < 1 .$$

The matrix $\boldsymbol{\Phi}$ is full column rank, and $\phi_\infty = \sqrt{2}$. Moreover, a direct calculation yields

$$\boldsymbol{\Phi}^\top \boldsymbol{D}\boldsymbol{\Phi} = \begin{bmatrix} \epsilon^2 & 0 \\ 0 & 1 \end{bmatrix} .$$

Consequently, $\omega = \lambda_{\min}\big(\boldsymbol{\Phi}^\top \boldsymbol{D}\boldsymbol{\Phi}\big) = \epsilon^2$. By choosing $\epsilon$ arbitrarily small, the curvature parameter $\omega$ can be made arbitrarily close to zero. Importantly, this degeneration arises solely from the feature representation: for different values of the mixing-rate parameter $\alpha$, the stationary distribution and $\omega$ remain unchanged.

## 5 Proof Sketch and Differences with Prior Proofs

The fact that the iterates are bounded in this explicit form is both a novel contribution and a crucial ingredient in our analysis. We now explain how this is obtained, contrasting it with previous approaches.

**Prior work: Controlling the iterates with curvature.** The standard approach to analyze TD(0) starts from the following recursion (Bhandari et al., 2018; Srikant & Ying, 2019; Sun et al., 2021; Patil et al., 2023; Mitra, 2024; Samsonov et al., 2024; Li et al., 2025):

$$\|\boldsymbol{\theta}_t - \boldsymbol{\theta}^*\|^2 = \|\boldsymbol{\theta}_{t-1} + \eta_{t-1}\boldsymbol{g}_{t-1} - \boldsymbol{\theta}^*\|^2 = \|\boldsymbol{\theta}_{t-1} - \boldsymbol{\theta}^*\|^2 + 2\eta_{t-1}\langle \boldsymbol{g}_{t-1}, \boldsymbol{\theta}_{t-1} - \boldsymbol{\theta}^*\rangle + \eta_{t-1}^2 \|\boldsymbol{g}_{t-1}\|^2$$
$$= \|\boldsymbol{\theta}_{t-1} - \boldsymbol{\theta}^*\|^2 + 2\eta_{t-1}\langle \bar{\boldsymbol{g}}(\boldsymbol{\theta}_{t-1}), \boldsymbol{\theta}_{t-1} - \boldsymbol{\theta}^*\rangle + \eta_{t-1}^2 \|\boldsymbol{g}_{t-1}\|^2 + 2\eta_{t-1}\langle \boldsymbol{g}_{t-1} - \bar{\boldsymbol{g}}(\boldsymbol{\theta}_{t-1}), \boldsymbol{\theta}_{t-1} - \boldsymbol{\theta}^*\rangle .$$

Then, in the fast regime (Srikant & Ying, 2019; Patil et al., 2023; Samsonov et al., 2024; Li et al., 2025), one can use the following lemma:

**Lemma 5.1.** *(Mitra, 2024, Lemma 1)*

$$\langle \bar{\boldsymbol{g}}(\boldsymbol{\theta}), \boldsymbol{\theta} - \boldsymbol{\theta}^*\rangle \leq -\omega(1-\gamma)\|\boldsymbol{\theta} - \boldsymbol{\theta}^*\|^2, \quad \forall \boldsymbol{\theta} \in \mathbb{R}^d.$$

Then, plugging this into the recursion yields

$$\|\boldsymbol{\theta}_t - \boldsymbol{\theta}^*\|^2 \leq (1 - 2\eta_{t-1}\omega(1-\gamma))\|\boldsymbol{\theta}_{t-1} - \boldsymbol{\theta}^*\|^2 + \eta_{t-1}^2 \|\boldsymbol{g}_{t-1}\|^2 + 2\eta_{t-1}\langle \boldsymbol{g}_{t-1} - \bar{\boldsymbol{g}}(\boldsymbol{\theta}_{t-1}), \boldsymbol{\theta}_{t-1} - \boldsymbol{\theta}^*\rangle .$$

One then proceeds to control the gradient term $\|\boldsymbol{g}_{t-1}\|^2$ and bias term $\langle \boldsymbol{g}_{t-1} - \bar{\boldsymbol{g}}(\boldsymbol{\theta}_{t-1}), \boldsymbol{\theta}_{t-1} - \boldsymbol{\theta}^*\rangle$ to form the following standard pseudo-contraction:

$$\mathbb{E}\left[\|\boldsymbol{\theta}_t - \boldsymbol{\theta}^*\|^2\right] = (1 - 2\eta_{t-1}\omega(1-\gamma))\mathbb{E}\left[\|\boldsymbol{\theta}_{t-1} - \boldsymbol{\theta}^*\|^2\right] + \mathcal{O}\left(\eta_{t-1}^2 \|\boldsymbol{\theta}^*\|^2\right),$$

Unrolling the recursion yields $\mathbb{E}\left[\left\|\boldsymbol{V}_{\boldsymbol{\theta}^*} - \boldsymbol{V}_{\bar{\boldsymbol{\theta}}_T}\right\|_{\boldsymbol{D}}^2\right] = \widetilde{\mathcal{O}}(\frac{\|\boldsymbol{\theta}^*\|^2}{\omega^2(1-\gamma)^2 T})$. Notice that there is no need for projection steps in this line of analysis, thanks to the contraction property, which keeps the iterates bounded. However, in this approach, the rate depends on $\omega$, which could be arbitrarily small, as we show in Section 4.1.

**Prior Work: Controlling the iterates with projections.** To avoid the dependency on $\omega$, we can analyze TD(0) using Lemma 4.1 instead of Lemma 5.1. The analysis starts from controlling the magnitude of the gradient $\|\boldsymbol{g}_{t-1}\|$ using the following lemma:

**Lemma 5.2.** *(Bhandari et al., 2018, Lemma 6)* For $Z_t = (s_t, s_{t+1})$, for all $\boldsymbol{\theta} \in \mathbb{R}^d$,

$$\|\boldsymbol{g}(\boldsymbol{\theta}, Z_t)\| \leq r_\infty \phi_\infty + 2\phi_\infty^2 \|\boldsymbol{\theta}\| .$$

Hence, the term $\|\boldsymbol{\theta}_{t-1}\|$ upper bounds the magnitude of the gradient $\|\boldsymbol{g}_{t-1}\|$. Since $\boldsymbol{\theta}_{t-1} = \boldsymbol{\theta}_{t-2} + \eta_{t-2}\boldsymbol{g}_{t-2}$, the term $\|\boldsymbol{\theta}_{t-1}\|$ in turn depends on both $\|\boldsymbol{\theta}_{t-2}\|$ and $\|\boldsymbol{g}_{t-2}\|$. This recursive dependence can create a vicious cycle, leading to an explosion in $\|\boldsymbol{\theta}_t\|$ driven by the stochasticity of $\boldsymbol{g}_{t-1}$. Previous analyses (Bhandari et al., 2018; Liu & Olshevsky, 2021) avoid this problem by imposing an artificial projection step, which guarantees $\|\boldsymbol{\theta}_t\| \leq R$ for all $t$, where $R$ is chosen agnostically to be larger than $\|\boldsymbol{\theta}^*\|$. Under this constraint, we have a uniform control over the magnitude of $\|\boldsymbol{g}_t\|$ by $r_\infty \phi_\infty + 2\phi_\infty^2 R$ for all $t$, which can be seen as a bounded gradients condition in optimization.

**Our Approach: Controlling the iterates *without* projections.** Now, we explain our proof method, which eliminates the need for a projection.

Let's first give our argument in a nutshell. We first ignore the presence of Markovian noise. Next, assuming the iterates are bounded up to time $t-1$, the next TD learning update is also bounded; this allows us to show that the next iterate remains bounded for a carefully chosen stepsize. This implication naturally suggests an inductive proof. For previously ignored Markovian noise, our strategy is to use the geometric convergence in Theorem 3.2. Thus, we work with the stationary update field $\bar{\boldsymbol{g}}$ and control the associated error term. By combining all these steps, we obtain the stated result.

Let's now look at the details. We decompose the updates as follows:

$$\boldsymbol{\theta}_t = \boldsymbol{\theta}_{t-1} + \eta_{t-1}\boldsymbol{g}_{t-1} = \boldsymbol{\theta}_{t-1} + \eta_{t-1}\left(\boldsymbol{g}_{t-1} - \mathbb{E}[\boldsymbol{g}_{t-1} \mid \mathcal{F}_{t-2}]\right) + \eta_{t-1}\left(\mathbb{E}[\boldsymbol{g}_{t-1} \mid \mathcal{F}_{t-2}] - \bar{\boldsymbol{g}}(\boldsymbol{\theta}_{t-1}) + \bar{\boldsymbol{g}}(\boldsymbol{\theta}_{t-1})\right).$$

Notice that $\boldsymbol{\xi}_{t-1} := \boldsymbol{g}_{t-1} - \mathbb{E}[\boldsymbol{g}_{t-1} \mid \mathcal{F}_{t-2}]$ is a martingale difference sequence with respect to $\mathcal{F}_{t-1} := \sigma(s_0, \ldots, s_t)$, and $\boldsymbol{b}_{t-1} := \mathbb{E}[\boldsymbol{g}_{t-1} \mid \mathcal{F}_{t-2}] - \bar{\boldsymbol{g}}(\boldsymbol{\theta}_{t-1})$ is the gradient bias term. Then, we have

$$
\begin{aligned}
\|\boldsymbol{\theta}_t - \boldsymbol{\theta}^*\|^2 &= \|\boldsymbol{\theta}_{t-1} + \eta_{t-1}(\boldsymbol{\xi}_{t-1} + \boldsymbol{b}_{t-1} + \bar{\boldsymbol{g}}(\boldsymbol{\theta}_{t-1})) - \boldsymbol{\theta}^*\|^2 \\
&= \|\boldsymbol{\theta}_{t-1} - \boldsymbol{\theta}^*\|^2 + 2\eta_{t-1}\langle \boldsymbol{\xi}_{t-1} + \boldsymbol{b}_{t-1} + \bar{\boldsymbol{g}}(\boldsymbol{\theta}_{t-1}), \boldsymbol{\theta}_{t-1} - \boldsymbol{\theta}^*\rangle \\
&\leq \|\boldsymbol{\theta}_{t-1} - \boldsymbol{\theta}^*\|^2 + 2\eta_{t-1}\langle \boldsymbol{\xi}_{t-1} + \boldsymbol{b}_{t-1}, \boldsymbol{\theta}_{t-1} - \boldsymbol{\theta}^*\rangle + 3\eta_{t-1}^2 \|\boldsymbol{\xi}_{t-1}\|^2 + 3\eta_{t-1}^2 \|\boldsymbol{b}_{t-1}\|^2 + 3\eta_{t-1}^2 \|\bar{\boldsymbol{g}}(\boldsymbol{\theta}_{t-1})\|^2,
\end{aligned}
$$

where we use Lemma 4.1 in the last inequality. Taking expectation and telescoping gives

$$
\mathbb{E}\left[\|\boldsymbol{\theta}_t - \boldsymbol{\theta}^*\|^2\right] \leq 2\mathbb{E}\left[\sum_{k=0}^{t-1} \eta_k \langle \boldsymbol{b}_k, \boldsymbol{\theta}_k - \boldsymbol{\theta}^*\rangle\right] + \|\boldsymbol{\theta}^*\|^2 + 3\mathbb{E}\left[\sum_{k=0}^{t-1} \eta_k^2(\|\boldsymbol{\xi}_k\|^2 + \|\boldsymbol{b}_k\|^2 + \|\bar{\boldsymbol{g}}(\boldsymbol{\theta}_k)\|^2)\right]. \tag{4}
$$

Our analysis follows from controlling the gradient bias term $\boldsymbol{b}_k$ in the update. The difficulty in analyzing $\boldsymbol{b}_k$ comes from $\boldsymbol{g}_k$ is not an unbiased estimate of $\bar{\boldsymbol{g}}(\boldsymbol{\theta}_k)$.

To be more precise, recall $Z_k := (s_k, s_{k+1})$. Using the TD update map $\boldsymbol{g}$ defined in (1), in general we have

$$
\mathbb{E}[\boldsymbol{g}(\boldsymbol{\theta}_k, Z_k) \mid \mathcal{F}_{k-1}] \neq \bar{\boldsymbol{g}}(\boldsymbol{\theta}_k).
$$

This is because $Z_k$ depends on $Z_{k-1} \in \mathcal{F}_{k-1}$, and the conditional law $\mathcal{L}(Z_k \mid Z_{k-1})$ is not necessarily equal to the stationary law.

The key idea for decoupling this dependency is that it converges to the stationary distribution geometrically fast. So, for large $k'$, the conditional law $\mathcal{L}(Z_k \mid Z_{k-k'})$ is close to the stationary law. This is characterized by the following lemma, whose proof is in Appendix D.

**Lemma 5.3.** *For any $0 \leq k' \leq k$, let $\ell_{k-k'} := r_\infty \phi_\infty + 2\phi_\infty^2 \|\boldsymbol{\theta}_{k-k'}\|$, then, we have with probability 1,*

$$
\|\mathbb{E}[\boldsymbol{g}(\boldsymbol{\theta}_{k-k'}, Z_k) - \bar{\boldsymbol{g}}(\boldsymbol{\theta}_{k-k'}) \mid \mathcal{F}_{k-k'-1}]\| \leq 2\ell_{k-k'} C\alpha^{k'}.
$$

Thus, we can decompose the scalar bias term appearing in (4). For any $0 \leq k' \leq k$, we have

$$
\begin{aligned}
\mathbb{E}[\langle \boldsymbol{b}_k, \boldsymbol{\theta}_k - \boldsymbol{\theta}^*\rangle] &= \mathbb{E}[\langle \mathbb{E}[\boldsymbol{g}(\boldsymbol{\theta}_{k-k'}, Z_k) \mid \mathcal{F}_{k-1}] - \bar{\boldsymbol{g}}(\boldsymbol{\theta}_{k-k'}), \boldsymbol{\theta}_{k-k'} - \boldsymbol{\theta}^*\rangle] \\
&\quad + \mathbb{E}[\langle \mathbb{E}[\boldsymbol{g}(\boldsymbol{\theta}_k, Z_k) \mid \mathcal{F}_{k-1}] - \bar{\boldsymbol{g}}(\boldsymbol{\theta}_k), \boldsymbol{\theta}_k - \boldsymbol{\theta}^*\rangle] \\
&\quad - \mathbb{E}[\langle \mathbb{E}[\boldsymbol{g}(\boldsymbol{\theta}_{k-k'}, Z_k) \mid \mathcal{F}_{k-1}] - \bar{\boldsymbol{g}}(\boldsymbol{\theta}_{k-k'}), \boldsymbol{\theta}_{k-k'} - \boldsymbol{\theta}^*\rangle].
\end{aligned}
$$

The first term is where Lemma 5.3 is applied. Since $\boldsymbol{\theta}_{k-k'} - \boldsymbol{\theta}^*$ is $\mathcal{F}_{k-k'-1}$-measurable and $\mathcal{F}_{k-k'-1} \subseteq \mathcal{F}_{k-1}$, the conditional expectation property gives

$$
\begin{aligned}
\mathbb{E}[\langle \mathbb{E}[\boldsymbol{g}(\boldsymbol{\theta}_{k-k'}, Z_k) \mid \mathcal{F}_{k-1}] &- \bar{\boldsymbol{g}}(\boldsymbol{\theta}_{k-k'}), \boldsymbol{\theta}_{k-k'} - \boldsymbol{\theta}^*\rangle] \\
&= \mathbb{E}[\langle \mathbb{E}[\boldsymbol{g}(\boldsymbol{\theta}_{k-k'}, Z_k) - \bar{\boldsymbol{g}}(\boldsymbol{\theta}_{k-k'}) \mid \mathcal{F}_{k-1}], \boldsymbol{\theta}_{k-k'} - \boldsymbol{\theta}^*\rangle] \\
&= \mathbb{E}[\langle \mathbb{E}[\boldsymbol{g}(\boldsymbol{\theta}_{k-k'}, Z_k) - \bar{\boldsymbol{g}}(\boldsymbol{\theta}_{k-k'}) \mid \mathcal{F}_{k-k'-1}], \boldsymbol{\theta}_{k-k'} - \boldsymbol{\theta}^*\rangle] \\
&\leq \mathbb{E}[\|\boldsymbol{\theta}_{k-k'} - \boldsymbol{\theta}^*\| \|\mathbb{E}[\boldsymbol{g}(\boldsymbol{\theta}_{k-k'}, Z_k) - \bar{\boldsymbol{g}}(\boldsymbol{\theta}_{k-k'}) \mid \mathcal{F}_{k-k'-1}]\|] \\
&\leq 2C\alpha^{k'}\mathbb{E}[\|\boldsymbol{\theta}_{k-k'} - \boldsymbol{\theta}^*\| \ell_{k-k'}]. \tag{5}
\end{aligned}
$$

Here, the second equality follows from the fact that $\boldsymbol{\theta}_{k-k'} - \boldsymbol{\theta}^*$ is $\mathcal{F}_{k-k'-1}$-measurable; equivalently, for this inner product, we may move the outer conditional expectation from $\mathcal{F}_{k-1}$ back to $\mathcal{F}_{k-k'-1}$. The first inequality uses Cauchy's inequality, and the second uses Lemma 5.3.

It remains to control the difference between the current scalar bias term and the delayed scalar bias term using the following lemma, whose proof is also in Appendix D.

**Lemma 5.4.** *Fix any $0 \leq k' \leq k \leq T-1$, then for any $\boldsymbol{\theta}_k$, $\boldsymbol{\theta}_{k'}$ and $Z_k$, if we assume $\sup_o \|\boldsymbol{g}(\boldsymbol{\theta}_k, o)\| \leq \ell_k$, we have*

$$
|\Xi(\boldsymbol{\theta}_k, Z_k) - \Xi(\boldsymbol{\theta}_{k-k'}, Z_k)| \leq (2\ell_k + 4\phi_\infty^2 \|\boldsymbol{\theta}_{k-k'} - \boldsymbol{\theta}^*\|) \|\boldsymbol{\theta}_k - \boldsymbol{\theta}_{k-k'}\|,
$$

*where $\Xi(\boldsymbol{\theta}, Z_t) := \langle \mathbb{E}[\boldsymbol{g}(\boldsymbol{\theta}, Z_t) \mid \mathcal{F}_{t-1}] - \bar{\boldsymbol{g}}(\boldsymbol{\theta}), \boldsymbol{\theta} - \boldsymbol{\theta}^*\rangle$.*

A comprehensive proof for controlling gradient bias terms for all $k \leq t$ can be found in Appendix F.1. Combining the scalar bound (5) with Lemma 5.4, we have

$$\mathbb{E}\left[\sum_{k=0}^{t-1} \eta_k \langle \boldsymbol{b}_k, \boldsymbol{\theta}_k - \boldsymbol{\theta}^* \rangle\right] = \mathcal{O}\left(\max_{i \leq t-1} \mathbb{E}\left[\|\boldsymbol{\theta}_i\|^2\right] + \|\boldsymbol{\theta}^*\|^2\right).$$

Plugging this into equation (4) and controlling the other gradient-like terms with Lemma 5.2, we conclude that $\mathbb{E}\left[\|\boldsymbol{\theta}_t - \boldsymbol{\theta}^*\|^2\right]$ is also of order

$$\mathcal{O}\left(\max_{i \leq t-1} \mathbb{E}\left[\|\boldsymbol{\theta}_i\|^2\right] + \|\boldsymbol{\theta}^*\|^2\right).$$

Notice that $\mathbb{E}\left[\|\boldsymbol{\theta}_t\|^2\right] \leq \mathbb{E}\left[(\|\boldsymbol{\theta}_t - \boldsymbol{\theta}^*\| + \|\boldsymbol{\theta}^*\|)^2\right]$ by the triangle inequality. That is, to bound $\mathbb{E}\left[\|\boldsymbol{\theta}_t\|^2\right]$, we can first bound the surrogate target $\mathbb{E}\left[\|\boldsymbol{\theta}_t - \boldsymbol{\theta}^*\|^2\right]$ using $\max_{i \leq t-1} \mathbb{E}\left[\|\boldsymbol{\theta}_i\|^2\right]$.

Motivated by this, we consider the induction hypothesis:

$$\max_{i \leq t-1} \mathbb{E}\left[\|\boldsymbol{\theta}_i\|^2\right] \leq \rho_c^2 \max\left\{\frac{r_\infty^2}{\phi_\infty^2}, \|\boldsymbol{\theta}^*\|^2\right\}$$

and prove that $\max_{i \leq t} \mathbb{E}[\|\boldsymbol{\theta}_i\|^2] \leq \rho_c^2 \max\left\{\frac{r_\infty^2}{\phi_\infty^2}, \|\boldsymbol{\theta}^*\|^2\right\}$. Indeed, if the stepsize parameter $c$ is large enough, a constant that bounds the previous iterates will also bound the next iterate. We remark again that the stepsize parameter $c$ and $\rho_c$ in induction are chosen carefully to ensure the inductive step proceeds. The precise derivation can be found in Theorem G.1 in the Appendix.

**Convergence result.** We now give the proof of Corollary 4.3.

*Proof.* For any $0 \leq t \leq T - 1$, let $d_t = \|\boldsymbol{\theta}^* - \boldsymbol{\theta}_t\|$. Thus,

$$d_{t+1}^2 = \|\boldsymbol{\theta}^* - \boldsymbol{\theta}_t - \eta_t \boldsymbol{g}_t\|^2 = d_t^2 - 2\eta_t \langle \boldsymbol{g}_t, \boldsymbol{\theta}^* - \boldsymbol{\theta}_t \rangle + \eta_t^2 \|\boldsymbol{g}_t\|^2$$
$$= d_t^2 - 2\eta_t \langle \bar{\boldsymbol{g}}(\boldsymbol{\theta}_t), \boldsymbol{\theta}^* - \boldsymbol{\theta}_t \rangle + 2\eta_t \langle \bar{\boldsymbol{g}}(\boldsymbol{\theta}_t) - \boldsymbol{g}_t, \boldsymbol{\theta}^* - \boldsymbol{\theta}_t \rangle + \eta_t^2 \|\boldsymbol{g}_t\|^2 \ .$$

Summing from $t = 0$ to $t = T - 1$, taking the expectation, and using Lemma 4.1, we have

$$\sum_{t=0}^{T-1} 2\eta_t \mathbb{E}\left[(1 - \gamma) \|\boldsymbol{V}_{\boldsymbol{\theta}_t} - \boldsymbol{V}_{\boldsymbol{\theta}^*}\|_{\boldsymbol{D}}^2 + \gamma \|\boldsymbol{V}_{\boldsymbol{\theta}_t} - \boldsymbol{V}_{\boldsymbol{\theta}^*}\|_{\text{Dir}}^2\right]$$
$$\leq \sum_{t=0}^{T-1} \left(\mathbb{E}[d_t^2] - \mathbb{E}[d_{t+1}^2]\right) + \mathbb{E}\left[\sum_{t=0}^{T-1} \eta_t^2 \|\boldsymbol{g}_t\|^2\right] + \mathbb{E}\left[\sum_{t=0}^{T-1} 2\eta_t \langle \bar{\boldsymbol{g}}(\boldsymbol{\theta}_t) - \boldsymbol{g}_t, \boldsymbol{\theta}^* - \boldsymbol{\theta}_t \rangle\right]$$
$$= \|\boldsymbol{\theta}^*\|^2 + \mathcal{O}\left(\rho_c^2 \max\left\{\frac{r_\infty^2}{\phi_\infty^2}, \|\boldsymbol{\theta}^*\|^2\right\}\right),$$

where we used Theorem 4.2 to upper bound $\mathbb{E}\left[\|\boldsymbol{\theta}_t\|^2\right]$ related terms by $\rho_c^2 \max\left\{\frac{r_\infty^2}{\phi_\infty^2}, \|\boldsymbol{\theta}^*\|^2\right\}$ in the last inequality. Using the convexity of $f$, we have

$$\mathbb{E}\left[f(\bar{\boldsymbol{\theta}}_T) - f(\boldsymbol{\theta}^*)\right] \leq \frac{1}{\sum_{i=0}^{T-1} \eta_i} \sum_{t=0}^{T-1} \eta_t \mathbb{E}\left[(1 - \gamma) \|\boldsymbol{V}_{\boldsymbol{\theta}_t} - \boldsymbol{V}_{\boldsymbol{\theta}^*}\|_{\boldsymbol{D}}^2\right] + \frac{1}{\sum_{i=0}^{T-1} \eta_i} \sum_{t=0}^{T-1} \eta_t \mathbb{E}\left[\gamma \|\boldsymbol{V}_{\boldsymbol{\theta}_t} - \boldsymbol{V}_{\boldsymbol{\theta}^*}\|_{\text{Dir}}^2\right]$$
$$= \widetilde{\mathcal{O}}\left(\frac{c\rho_c^2 \max\{r_\infty^2, \phi_\infty^2 \|\boldsymbol{\theta}^*\|^2\}}{\sqrt{T}}\right),$$

where we use $\sum_{i=0}^{T-1} \eta_i \geq \frac{2\sqrt{T+1} - 2}{c\phi_\infty^2 \log^2(T+3)}$ in the last equality. $\qquad\square$

# 6 Conclusion

In this paper, we present a robust finite-time analysis of TD(0) without requiring additional projection steps. To the best of our knowledge, this is the first finite-time guarantee in this setting. In particular, we do not employ the contraction-based proof technique used in previous work; instead, we directly prove that the iterates of TD(0) are bounded. We believe our proof is general and, for example, it can be easily extended to analyze the TD($\lambda$) algorithm and the Q-learning setting introduced in (Chen et al., 2022).

In future work, we plan to investigate the possibility of obtaining rates that interpolate between $\widetilde{\mathcal{O}}(1/\sqrt{T})$ and $\widetilde{\mathcal{O}}(1/T)$, depending on the curvature of the potential function. Ideally, one would like to show that TD(0) adapts to the curvature of the function with a specific stepsize, as is possible with recent parameter-free schemes (Cutkosky & Orabona, 2018).

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

# A Evidence on Complementarity of Fast and Robust Rates

The fact that fast and robust rates are complementary and both worth studying is well known among experts in stochastic approximation. However, because people rarely state obvious facts, younger generations may sometimes miss them. Hence, to help a reader who might have missed such subtlety, here we report quotes from seminal papers on the topic of fast vs. robust rates.

- Bhandari et al. (2018), Section 8.1, on the complementarity of the robust and fast rate: "As before, in the spirit of robust stochastic approximation [Nemirovski et al., 2009], the bound in part (a) gives a comparatively slow convergence rate of $\widetilde{\mathcal{O}}(1/\sqrt{T})$, but where the bound and step-size sequence are independent of the conditioning of the feature covariance matrix $\boldsymbol{\Sigma}$. The bound in part (c) gives a faster convergence rate in terms of the number of samples $T$, but the bound and as well as the step-size sequence depend on the minimum eigenvalue $\omega$ of $\boldsymbol{\Sigma}$."

- Samsonov et al. (2024), Section 3, on the comparison with robust approaches: "**Comparison to the robust SA approach.** Note that the leading term of the bound in Theorem 3 includes factors of $1/\lambda_{\min}$. This dependence is generally unavoidable if one aims to obtain the MSE bound for $\mathbb{E}[\|\bar{\boldsymbol{\theta}}_n - \boldsymbol{\theta}^*\|_{\boldsymbol{\Sigma}_\phi}^2]$ that scales as $1/n$. [...] In contrast, within the basin of robust stochastic approximation (RSA, (Nemirovski et al., 2009)), a convergence rate for $\mathbb{E}[\|\bar{\boldsymbol{\theta}}_n - \boldsymbol{\theta}^*\|_{\boldsymbol{\Sigma}_\phi}^2]$ of order $O(1/\sqrt{n})$ can be derived with the instance-independent choice of step size. Importantly, this rate is not affected by a worst-case factor of $\lambda_{\min}^{-1}$. This result was obtained for the TD algorithm in (Bhandari et al., 2018, Theorem 2)."

- Liu & Olshevsky (2021), Section 4.1, on the advantage of rates that are independent of the curvature: "One issue is the choice of step-size. The existing literature on temporal difference learning contains a range of possible step-sizes from $O(1/t)$ to $O(1/\sqrt{t})$ (see Bhandari et al. (2018); Dalal et al. (2018); Lakshminarayanan and Szepesvari (2018)). A step-size that scales as $O(1/\sqrt{t})$ is often preferred because, for faster decaying step-sizes, performance will scale with the smallest eigenvalue of $\boldsymbol{\Phi}^\top \boldsymbol{D} \boldsymbol{\Phi}$ or related quantity, and these can be quite small. This is not the case, however, for a step-size that decays like $O(1/\sqrt{t})$."

*Proof.* For any $0 \leq t \leq T - 1$, let $d_t = \|\boldsymbol{\theta}^* - \boldsymbol{\theta}_t\|$. Thus,

$$d_{t+1}^2 = \|\boldsymbol{\theta}^* - \boldsymbol{\theta}_t - \eta_t \boldsymbol{g}_t\|^2 = d_t^2 - 2\eta_t \langle \boldsymbol{g}_t, \boldsymbol{\theta}^* - \boldsymbol{\theta}_t \rangle + \eta_t^2 \|\boldsymbol{g}_t\|^2$$
$$= d_t^2 - 2\eta_t \langle \bar{\boldsymbol{g}}(\boldsymbol{\theta}_t), \boldsymbol{\theta}^* - \boldsymbol{\theta}_t \rangle + 2\eta_t \langle \bar{\boldsymbol{g}}(\boldsymbol{\theta}_t) - \boldsymbol{g}_t, \boldsymbol{\theta}^* - \boldsymbol{\theta}_t \rangle + \eta_t^2 \|\boldsymbol{g}_t\|^2 .$$

Summing from $t = 0$ to $t = T - 1$, taking the expectation, and using Lemma 4.1, we have

$$\sum_{t=0}^{T-1} 2\eta_t \mathbb{E}\Big[(1 - \gamma) \|\boldsymbol{V}_{\boldsymbol{\theta}_t} - \boldsymbol{V}_{\boldsymbol{\theta}^*}\|_{\boldsymbol{D}}^2 + \gamma \|\boldsymbol{V}_{\boldsymbol{\theta}_t} - \boldsymbol{V}_{\boldsymbol{\theta}^*}\|_{\text{Dir}}^2\Big]$$

$$\leq \sum_{t=0}^{T-1} \left(\mathbb{E}[d_t^2] - \mathbb{E}[d_{t+1}^2]\right) + \mathbb{E}\left[\sum_{t=0}^{T-1} \eta_t^2 \|\boldsymbol{g}_t\|^2\right] + \mathbb{E}\left[\sum_{t=0}^{T-1} 2\eta_t \langle \bar{\boldsymbol{g}}(\boldsymbol{\theta}_t) - \boldsymbol{g}_t, \boldsymbol{\theta}^* - \boldsymbol{\theta}_t \rangle\right]$$

$$= \|\boldsymbol{\theta}^*\|^2 + \mathcal{O}\left(\rho_c^2 \max\left\{\frac{r_\infty^2}{\phi_\infty^2}, \|\boldsymbol{\theta}^*\|^2\right\}\right),$$

where we used Theorem G.1 in the last inequality. Using the convexity of $f$, we have

$$\mathbb{E}\left[f(\bar{\boldsymbol{\theta}}_T) - f(\boldsymbol{\theta}^*)\right] \leq \frac{1}{\sum_{i=0}^{T-1} \eta_i} \sum_{t=0}^{T-1} \eta_t \mathbb{E}\Big[(1 - \gamma) \|\boldsymbol{V}_{\boldsymbol{\theta}_t} - \boldsymbol{V}_{\boldsymbol{\theta}^*}\|_{\boldsymbol{D}}^2\Big] + \frac{1}{\sum_{i=0}^{T-1} \eta_i} \sum_{t=0}^{T-1} \eta_t \mathbb{E}\Big[\gamma \|\boldsymbol{V}_{\boldsymbol{\theta}_t} - \boldsymbol{V}_{\boldsymbol{\theta}^*}\|_{\text{Dir}}^2\Big]$$

$$= \widetilde{\mathcal{O}}\left(\frac{c\rho_c^2 \max\{r_\infty^2, \phi_\infty^2 \|\boldsymbol{\theta}^*\|^2\}}{\sqrt{T}}\right),$$

where we use $\sum_{i=0}^{T-1} \eta_i \geq \frac{2\sqrt{T+1} - 2}{c\phi_\infty^2 \log^2(T+3)}$ in the last equality. $\qquad\square$

## B  On the Inputs of TD Learning Algorithms

In this section, we discuss the hyperparameter requirements of various TD learning algorithms to achieve rates in Table 1.

We begin with the fast regime.

In Bhandari et al. (2018, Theorem 3(c)), with $\phi_\infty^2 = 1$. The stepsize is chosen as

$$\eta_t = \frac{1}{\omega(t+1)(1-\gamma)}.$$

So prior knowledge of $\phi_\infty$ and $\omega$ is required to determine the step sizes. However, their rate depends on $\tau(\eta_T)$. Thus, in order to achieve the true $\widetilde{\mathcal{O}}(1/T)$ rate (eliminating the dependency on $\alpha$), the knowledge of $\omega$, $\phi_\infty$, $\alpha$ and $T$ is required.

In Srikant & Ying (2019, Theorem 7), the constant stepsize $\eta$ is chosen to satisfy

$$\frac{32}{\omega}(1 + r_\infty \phi_\infty + 2\phi_\infty^2 \|\boldsymbol{\theta}^*\|)\eta\tau(\eta) + \frac{\eta}{2\omega} \le 0.05$$

So the prior knowledge of $\phi_\infty$, $\alpha$, $\omega$ and $\|\boldsymbol{\theta}^*\|$ is required for implementing the algorithm.

In Patil et al. (2023, Section 7), the stepsize is set to the constant value

$$\eta = \frac{1-\gamma}{(1+\gamma)^2 \phi_\infty^2} \ ,$$

and the data-dropping block length can be chosen as $K = \frac{1}{\log(1/\alpha)} \log(CT^3)$ with a fixed $T$ and $\delta = \frac{1}{T^2}$. Consequently, implementing the algorithm requires prior knowledge of $\phi_\infty$ to set the stepsize, and $\alpha$ to set the block length.

In Samsonov et al. (2024, Theorem 6), the stepsize can be set to the constant value

$$\eta = \frac{1-\gamma}{384 \log T} \ ,$$

under the assumption that $\phi_\infty^2 = 1$ and choosing $\delta = \frac{1}{T^2}$. The data-dropping block length can also be chosen as $q = \lceil \frac{3 \log T}{\log(1/\alpha) \log 4} \rceil$. Consequently, implementing the algorithm requires prior knowledge of $\phi_\infty$ to set the stepsize, and $\alpha$ to set the block length.

In Mitra (2024, Theorem 1), the constant stepsize $\eta$ is chosen to satisfy

$$\eta \le \frac{\omega(1-\gamma)}{C_1 \tau(\eta)} \ ,$$

for some universal constant $C_1 \ge 8$ under the assumption $\phi_\infty^2 = 1$. So the prior knowledge of $\phi_\infty$, $\alpha$, and $\omega$ is required for implementing the algorithm.

In Li et al. (2025, Theorem 4.10), the exponential stepsize $\eta_t$ is defined as

$$\eta_t := \eta_0 \left(\frac{1}{T}\right)^{t/T}$$

under the assumption $\phi_\infty^2 = 1$ and $\eta_0$ depends on $\omega$. Notably, $T$ needs to be set large enough to compensate for $\alpha$.

Let's now move to the robust regime.

In Bhandari et al. (2018, Theorem 3(a)) and Liu & Olshevsky (2021, Corollary 2), the stepsize is chosen as

$$\eta_t = 1/\sqrt{T}$$

under the assumption $\phi_\infty^2 = 1$. So running the algorithm requires only the knowledge of $\phi_\infty$. However, as proved in Appendix I, to make their rates independent of $\alpha$ and of the order $\widetilde{\mathcal{O}}(1/\sqrt{T})$. $T$ needs to be set large enough depending on $\alpha$.

In Sun et al. (2021, Theorem 3(a)) the stepsize is chosen as

$$\eta > 0$$

under the assumption $\phi_\infty^2 = 1$. Thus, running the algorithm requires only the knowledge of $\phi_\infty$. However, to obtain the $\widetilde{\mathcal{O}}(1/\sqrt{T})$ rate, $\eta$ would need to be set with prior knowledge of $\alpha$, $\omega$, $\phi_\infty$ and $T$.

In our work, the stepsize is chosen as

$$\eta_t = \frac{1}{c\phi_\infty^2 \log(T)\sqrt{t+1}\log(t+3)}.$$

We remark that $T$ is large enough compared to $\alpha$.

## C   Summary of Notation

We will assume $|r(s,s')| \leq r_\infty$ and $\|\boldsymbol{\phi}(s)\| \leq \phi_\infty$ for all $s, s' \in \mathcal{S}$. For all $t \leq T$, we recall the following notation:

$$
\begin{aligned}
&\mathcal{F}_t := \sigma(s_0, \ldots, s_{t+1}), \mathcal{F}_{-1} := \sigma(s_0),\\
&Z_t := (s_t, s_{t+1}),\\
&d_t := \|\boldsymbol{\theta}_t - \boldsymbol{\theta}^*\| \in \mathcal{F}_{t-1}, d_0 := \|\boldsymbol{\theta}^*\|,\\
&\ell_t := r_\infty \phi_\infty + 2\phi_\infty^2 \|\boldsymbol{\theta}_t\| \in \mathcal{F}_{t-1},\\
&\textcolor{red}{\boldsymbol{g}_t := \boldsymbol{g}(\boldsymbol{\theta}_t, Z_t) \in \mathcal{F}_t,}\\
&\boldsymbol{\xi}_t := \boldsymbol{g}(\boldsymbol{\theta}_t, Z_t) - \mathbb{E}[\boldsymbol{g}(\boldsymbol{\theta}_t, Z_t) \mid \mathcal{F}_{t-1}] \in \mathcal{F}_t,\\
&\boldsymbol{b}_t := \mathbb{E}[\boldsymbol{g}(\boldsymbol{\theta}_t, Z_t) \mid \mathcal{F}_{t-1}] - \bar{\boldsymbol{g}}(\boldsymbol{\theta}_t) \in \mathcal{F}_{t-1}.
\end{aligned}
$$

## D   Proof of Lemma 5.3 and Lemma 5.4

The proof of Lemma 5.3 is adapted from Bhandari et al. (2018, Lemma 9), and we state it here for completeness.

*Proof.* Recall the Markov chain $(s_t)_{t\geq 0}$ induced by the policy $\mu$, with transition probability $P^\mu$, stationary distribution $\pi$, and geometric mixing

$$\sup_{s \in \mathcal{S}}\left\|(P^\mu)^t(\cdot \mid s) - \pi\right\|_{\mathrm{TV}} \leq C\alpha^t, \qquad t \geq 0,$$

as stated in Theorem 3.2. Let $\mathcal{F}_t := \sigma(s_0, \ldots, s_{t+1})$ and $Z_t := (s_t, s_{t+1})$. Define

$$\bar{\boldsymbol{g}}(\boldsymbol{\theta}) := \mathbb{E}[\boldsymbol{g}(\boldsymbol{\theta}, Z)],$$

where $Z = (S, S')$ with $S \sim \pi$ and $S' \sim P^\mu(S, \cdot)$, and the pair $(S, S')$ is independent of algorithm's trajectory. By construction, $Z$ has the stationary law of the pair $(s_t, s_{t+1})$.

Fix integers $k$, $k'$ with $0 \leq k' \leq k$ and set $t := k - k'$. Note that, by the algorithmic recursion, $\boldsymbol{\theta}_t$ is $\mathcal{F}_{t-1}$-measurable, and $s_t$ is also $\mathcal{F}_{t-1}$-measurable since $Z_{t-1} = (s_{t-1}, s_t) \in \mathcal{F}_{t-1}$.

Let $(\Omega, \mathcal{G})$ be a measurable space, $P, Q$ probability measures on $(\Omega, \mathcal{G})$, and $f : \Omega \to \mathbb{R}^d$ a measurable function with

$$\|f\|_\infty := \sup_{\omega \in \Omega} \|f(\omega)\| < \infty.$$

We claim that

$$\left\|\mathbb{E}_P[f] - \mathbb{E}_Q[f]\right\| \leq 2\|f\|_\infty \, d_{\mathrm{TV}}(P, Q), \tag{6}$$

where $d_{\text{TV}}$ denotes total-variation distance.

Indeed, for any unit vector $u \in \mathbb{R}^d$ with $\|u\| = 1$, define the scalar function

$$h_u(\omega) := \frac{\langle u, f(\omega)\rangle}{2\|f\|_\infty}.$$

Then $|h_u(\omega)| \leq 1/2$ for all $\omega$. By the variational representation of total variation,

$$d_{\text{TV}}(P, Q) = \sup_{\|h\|_\infty \leq 1/2} \left|\mathbb{E}_P[h] - \mathbb{E}_Q[h]\right|.$$

Thus, for every such $u$,

$$\left|\mathbb{E}_P[h_u] - \mathbb{E}_Q[h_u]\right| \leq d_{\text{TV}}(P, Q),$$

and hence

$$\left|\langle u, \mathbb{E}_P[f] - \mathbb{E}_Q[f]\rangle\right| = 2\|f\|_\infty \left|\mathbb{E}_P[h_u] - \mathbb{E}_Q[h_u]\right| \leq 2\|f\|_\infty d_{\text{TV}}(P, Q).$$

Taking the supremum over all unit vectors $u$ yields

$$\left\|\mathbb{E}_P[f] - \mathbb{E}_Q[f]\right\| = \sup_{\|u\|=1}\left|\langle u, \mathbb{E}_P[f] - \mathbb{E}_Q[f]\rangle\right| \leq 2\|f\|_\infty d_{\text{TV}}(P, Q),$$

which proves (6).

Now fix a state $s \in \mathcal{S}$, and consider the Markov chain started from $s_t = s$. By time-homogeneity, the distribution of $s_k$ given $s_t = s$ is $(P^\mu)^{k'}(\cdot \mid s)$, and $s_{k+1}$ is then sampled from $P^\mu(\cdot \mid s_k)$.

Let $\nu_s$ be the law of the pair $(s_k, s_{k+1})$ given $s_t = s$, and let $\nu$ be the stationary law of $(S, S')$ defined above. We claim that

$$d_{\text{TV}}(\nu_s, \nu) \leq \left\|(P^\mu)^{k'}(\cdot \mid s) - \pi\right\|_{\text{TV}} \leq C\alpha^{k'}. \tag{7}$$

To see the first inequality, note that both $\nu_s$ and $\nu$ use the same conditional kernel $P^\mu$ for the second coordinate. For any measurable $A \subseteq \mathcal{S} \times \mathcal{S}$, write

$$A_x := \{y \in \mathcal{S} : (x, y) \in A\}.$$

Then

$$\nu_s(A) = \sum_{x \in \mathcal{S}}(P^\mu)^{k'}(s, x)P^\mu(x, A_x), \qquad \nu(A) = \sum_{x \in \mathcal{S}}\pi(x)P^\mu(x, A_x),$$

so

$$\nu_s(A) - \nu(A) = \sum_{x \in \mathcal{S}}\left((P^\mu)^{k'}(s, x) - \pi(x)\right)P^\mu(x, A_x).$$

Since $0 \leq P^\mu(x, A_x) \leq 1$, the supremum of $|\nu_s(A) - \nu(A)|$ over all measurable $A$ equals the total-variation distance between the first marginals, that is,

$$d_{\text{TV}}(\nu_s, \nu) = \left\|(P^\mu)^{k'}(\cdot \mid s) - \pi\right\|_{\text{TV}}.$$

By the mixing bound from Theorem 3.2, the right-hand side is at most $C\alpha^{k'}$, which proves (7).

Now we have all the tools to work on the original probability space and condition on the $\sigma$-field $\mathcal{F}_{t-1}$. For each outcome $\omega$, the realizations $\boldsymbol{\theta}_t(\omega)$ and $s_t(\omega)$ are fixed. By the Markov property and time-homogeneity, the conditional law of $Z_k = (s_k, s_{k+1})$ given $\mathcal{F}_{t-1}$ depends on $\omega$ only through $s_t(\omega)$, and coincides with $\nu_{s_t(\omega)}$:

$$\mathcal{L}\left(Z_k \mid \mathcal{F}_{t-1}\right)(\omega) = \nu_{s_t(\omega)}.$$

Let $\nu$ again denote the stationary law of $Z = (S, S')$, which does not depend on $\omega$. By (7), for every $\omega$,

$$d_{\text{TV}}\left(\mathcal{L}\left(Z_k \mid \mathcal{F}_{t-1}\right)(\omega), \nu\right) \leq C\alpha^{k'}.$$

Fix $\omega$ and set $\boldsymbol{\theta} := \boldsymbol{\theta}_t(\omega)$, and define a function $f_{\boldsymbol{\theta}} : \mathcal{S} \times \mathcal{S} \to \mathbb{R}^d$ by $f_{\boldsymbol{\theta}}(o) := \boldsymbol{g}(\boldsymbol{\theta}, o)$. Then, $\|f_{\boldsymbol{\theta}}\|_\infty = \sup_o \|\boldsymbol{g}(\boldsymbol{\theta}, o)\|$. Using (6) with $P = \mathcal{L}(Z_k \mid \mathcal{F}_{t-1})(\omega)$ and $Q = \nu$, we obtain

$$\left\| \mathbb{E}_P\big[\boldsymbol{g}(\boldsymbol{\theta}_t, o)\big] - \mathbb{E}_\nu\big[\boldsymbol{g}(\boldsymbol{\theta}_t, o)\big] \right\| \le 2 \sup_o \|\boldsymbol{g}(\boldsymbol{\theta}_t, o)\| \, C\alpha^{k'}.$$

Note that $Z$ with law $\nu$ is independent of $\mathcal{F}_{t-1}$, so

$$\mathbb{E}_\nu\big[\boldsymbol{g}(\boldsymbol{\theta}_t, o)\big] = \mathbb{E}\big[\boldsymbol{g}(\boldsymbol{\theta}_t, Z) \mid \mathcal{F}_{t-1}\big](\omega) = \bar{\boldsymbol{g}}(\boldsymbol{\theta}_t(\omega)).$$

And by construction,

$$\mathbb{E}_P\big[\boldsymbol{g}(\boldsymbol{\theta}_t, o)\big] = \mathbb{E}\big[\boldsymbol{g}(\boldsymbol{\theta}_t, Z_k) \mid \mathcal{F}_{t-1}\big](\omega).$$

Thus, for almost every $\omega$,

$$\left\| \mathbb{E}\big[\boldsymbol{g}(\boldsymbol{\theta}_t, Z_k) - \bar{\boldsymbol{g}}(\boldsymbol{\theta}_t) \mid \mathcal{F}_{t-1}\big](\omega) \right\| \le 2 \sup_o \|\boldsymbol{g}(\boldsymbol{\theta}_t, o)\| \, C\alpha^{k'}.$$

Recall that $t = k - k'$, so $\mathcal{F}_{t-1} = \mathcal{F}_{k-k'-1}$ and $\boldsymbol{\theta}_t = \boldsymbol{\theta}_{k-k'}$. Hence

$$\left\| \mathbb{E}[\boldsymbol{g}(\boldsymbol{\theta}_{k-k'}, Z_k) - \bar{\boldsymbol{g}}(\boldsymbol{\theta}_{k-k'}) \mid \mathcal{F}_{k-k'-1}] \right\| \le 2 \sup_o \|\boldsymbol{g}(\boldsymbol{\theta}_{k-k'}, o)\| \, C\alpha^{k'} \text{ a.s.}$$

which completes the proof. □

The proof of Lemma 5.4 is straightforward and we give it here for completeness.

*Proof.* We have

$$
\begin{aligned}
|\Xi(\boldsymbol{\theta}_k, Z_k) - \Xi(\boldsymbol{\theta}_{k'}, Z_k)| &= \Big| \langle \mathbb{E}[\boldsymbol{g}(\boldsymbol{\theta}_k, Z_k) \mid \mathcal{F}_{k-1}] - \bar{\boldsymbol{g}}(\boldsymbol{\theta}_k), \, \boldsymbol{\theta}_k - \boldsymbol{\theta}^* \rangle - \langle \mathbb{E}[\boldsymbol{g}(\boldsymbol{\theta}_{k'}, Z_k) \mid \mathcal{F}_{k-1}] - \bar{\boldsymbol{g}}(\boldsymbol{\theta}_{k'}), \, \boldsymbol{\theta}_{k'} - \boldsymbol{\theta}^* \rangle \Big| \\
&\le \left\| \mathbb{E}[\boldsymbol{g}(\boldsymbol{\theta}_k, Z_k) \mid \mathcal{F}_{k-1}] - \bar{\boldsymbol{g}}(\boldsymbol{\theta}_k) \right\| \|\boldsymbol{\theta}_k - \boldsymbol{\theta}_{k'}\| \\
&\quad + \|\boldsymbol{\theta}_{k'} - \boldsymbol{\theta}^*\| \left\| \big(\mathbb{E}[\boldsymbol{g}(\boldsymbol{\theta}_k, Z_k) \mid \mathcal{F}_{k-1}] - \bar{\boldsymbol{g}}(\boldsymbol{\theta}_k)\big) - \big(\mathbb{E}[\boldsymbol{g}(\boldsymbol{\theta}_{k'}, Z_k) \mid \mathcal{F}_{k-1}] - \bar{\boldsymbol{g}}(\boldsymbol{\theta}_{k'})\big) \right\| \\
&\le 2\ell_k \|\boldsymbol{\theta}_k - \boldsymbol{\theta}_{k'}\| + d_{k'}(2\phi_\infty^2 \|\boldsymbol{\theta}_k - \boldsymbol{\theta}_{k'}\| + 2\phi_\infty^2 \|\boldsymbol{\theta}_k - \boldsymbol{\theta}_{k'}\|) \\
&\le (2\ell_k + 4\phi_\infty^2 d_{k'}) \|\boldsymbol{\theta}_k - \boldsymbol{\theta}_{k'}\| \,,
\end{aligned}
$$

where we used Lemma E.5 with Jensen inequality in the second-to-last inequality. □

## E  Technical Lemmas

**Lemma E.1.** *Suppose that $0 \le u < t$. Then, we have*

$$\sum_{k=u+1}^{t-1} \frac{1}{\log(k+3)\log(k-u+3)\sqrt{k+1}\sqrt{k-u+1}} \le \frac{2}{\log 3} \,.$$

*Proof.* By re-indexing, we have

$$\sum_{k=u+1}^{t-1} \frac{1}{\log(k+3)\,\log(k-u+3)\,\sqrt{k+1}\,\sqrt{k-u+1}} \le \sum_{j=1}^{t-u-1} \frac{1}{(j+1)\,\log^2(j+3)} \,.$$

Consider the function

$$h(x) := \frac{1}{(x+1)\,\log^2(x+3)}, \qquad x \ge 1 \,.$$

For $x \ge 1$ we have $x + 1 \ge \frac{x+3}{2}$, hence

$$h(x) = \frac{1}{(x+1)\,\log^2(x+3)} \le \frac{2}{(x+3)\,\log^2(x+3)} \,.$$

Substituting $y = \log(x + 3)$ and using the fact that $h$ is decreasing, we have

$$\sum_{j=1}^{t-u-1} \frac{1}{(j+1)\log^2(j+3)} \leq \int_0^{t-u-1} \frac{2\mathrm{d}x}{(x+3)\log^2(x+3)} = \frac{2}{\log 3} - \frac{2}{\log(t-u+2)} \; . \qquad \square$$

**Lemma E.2.** *Suppose that $0 \leq u < t$. Then, we have*

$$\sum_{k=u+1}^{t-1} \frac{1}{\log(k+3)\sqrt{k+1}\sqrt{t}} \leq \frac{2}{\log(u+4)} \; .$$

*Proof.* We first factor out $\frac{1}{\sqrt{t}}$:

$$S_{u,t} = \frac{1}{\sqrt{t}} \sum_{k=u+1}^{t-1} \frac{1}{\log(k+3)\sqrt{k+1}} \; .$$

Because $k \geq u + 1$,

$$\log(k+3) \geq \log(u+4) \implies \frac{1}{\log(k+3)} \leq \frac{1}{\log(u+4)} \; .$$

Hence

$$S_{u,t} \leq \frac{1}{\sqrt{t}\log(u+4)} \sum_{k=u+1}^{t-1} \frac{1}{\sqrt{k+1}} \; .$$

The map $x \mapsto x^{-1/2}$ is positive and monotonically decreasing, so

$$\sum_{k=u+1}^{t-1} \frac{1}{\sqrt{k+1}} \leq \int_u^t x^{-\frac{1}{2}} \, dx = 2\left(\sqrt{t} - \sqrt{u}\right) \; .$$

Thus

$$S_{u,t} \leq \frac{2(\sqrt{t} - \sqrt{u})}{\sqrt{t}\log(u+4)} = \frac{2}{\log(u+4)}\left(1 - \sqrt{\frac{u}{t}}\right) \leq \frac{2}{\log(u+4)} \; . \qquad \square$$

**Lemma E.3.** *For every integer $t \geq 1$,*

$$\sum_{k=0}^{t-1} \frac{1}{\log^2(k+3)\,(k+1)} \leq \frac{1}{\log^2 3} + \frac{2}{\log 3} - \frac{2}{\log(t+2)} \leq \frac{1}{\log^2 3} + \frac{2}{\log 3} < 3 \; .$$

*Proof.* Denote the sum by $S_t$ and separate the $k = 0$ term:

$$S_t = \frac{1}{\log^2 3} + \sum_{k=1}^{t-1} \frac{1}{\log^2(k+3)(k+1)} \; .$$

For every $k \geq 1$ we have $k + 1 \geq \frac{1}{2}(k + 3)$, hence

$$\frac{1}{\log^2(k+3)(k+1)} \leq \frac{2}{\log^2(k+3)(k+3)} \; .$$

Because $x \mapsto \dfrac{1}{x\log^2 x}$ is positive and decreasing for $x \geq 3$,

$$\sum_{k=1}^{t-1} \frac{2}{\log^2(k+3)(k+3)} \leq 2\int_3^{t+2} \frac{dx}{x\log^2 x} \; .$$

The antiderivative is $-\dfrac{1}{\log x}$, so

$$2\int_{3}^{t+2}\frac{dx}{x\log^2 x} \;=\; 2\left[-\frac{1}{\log x}\right]_{3}^{t+2} \;=\; \frac{2}{\log 3}-\frac{2}{\log(t+2)}\;.$$

Thus,

$$S_t \;\le\; \frac{1}{\log^2 3}+\frac{2}{\log 3}-\frac{2}{\log(t+2)}\;. \hspace{2cm} \square$$

**Lemma E.4.** *Let $\alpha \in [1/2, 1)$ and define $L := \log(1/\alpha) > 0$. For $t \ge 1$, set $u_t = \left\lceil \frac{\log(2\sqrt{t})}{L} \right\rceil$. Assume that $T \in \mathbb{N}^+$ satisfies $\log T \ge \frac{2}{L^3}$. Then for every integer $t$ with $1 \le t \le T$,*

$$\log^2 T \;\ge\; (u_t + 1)^{3/2}.$$

*Proof.* Write $X := \log T$. Since $\lceil y \rceil \le y + 1$ for all real $y$, for any $1 \le t \le T$ we have

$$u_t + 1 = \left\lceil \frac{\log(2\sqrt{t})}{L} \right\rceil + 1 \le \frac{\log(2\sqrt{t})}{L} + 2 \le \frac{\log(2\sqrt{T})}{L} + 2.$$

Using $\log(2\sqrt{T}) = \frac{1}{2}\log T + \log 2 = \frac{1}{2}X + \log 2$, this becomes

$$u_t + 1 \le \frac{\frac{1}{2}X + \log 2}{L} + 2.$$

From the assumption $X \ge 2/L^3$ we get $X^{1/3} \ge 2^{1/3}/L$, hence

$$X^{4/3} = X \cdot X^{1/3} \;\ge\; 2^{1/3}\frac{X}{L}.$$

Therefore, it suffices to show

$$\frac{\frac{1}{2}X + \log 2}{L} + 2 \;\le\; 2^{1/3}\frac{X}{L},$$

or equivalently (multiplying by $L > 0$)

$$\left(2^{1/3} - \tfrac{1}{2}\right)X \;\ge\; \log 2 + 2L.$$

Because $\alpha \in [1/2, 1)$, we have $L = \log(1/\alpha) \le \log 2$, hence $\log 2 + 2L \le 3\log 2$. Also $X \ge 2/L^3 \ge 2/(\log 2)^3$, so it is enough to verify

$$\left(2^{1/3} - \tfrac{1}{2}\right)\frac{2}{(\log 2)^3} \;\ge\; 3\log 2,$$

which is equivalent to

$$2\left(2^{1/3} - \tfrac{1}{2}\right) \;\ge\; 3(\log 2)^4.$$

Now $\log 2 < 3/4$ since $e^{3/4} = 1 + \frac{3}{4} + \frac{(3/4)^2}{2} + \cdots > 1 + \frac{3}{4} + \frac{9}{32} = 2.03125 > 2$. Thus

$$3(\log 2)^4 < 3\left(\frac{3}{4}\right)^4 = \frac{243}{256} < 1.$$

On the other hand,

$$2\left(2^{1/3} - \tfrac{1}{2}\right) = 2^{4/3} - 1 > 2 - 1 = 1,$$

so indeed $2(2^{1/3} - 1/2) > 3(\log 2)^4$. This proves $u_t + 1 \le X^{4/3}$ for all $1 \le t \le T$.

Finally, raising both sides to the power $3/2$ yields

$$(u_t + 1)^{3/2} \le \left(X^{4/3}\right)^{3/2} = X^2 = (\log T)^2,$$

which is the desired inequality. $\hspace{2cm} \square$

**Lemma E.5.** *(Bhandari et al., 2018, Lemma 10) Fix any $k \leq T$, for any $Z_k$, $\boldsymbol{\theta}$ and $\boldsymbol{\theta}'$, we have*

$$\|\boldsymbol{g}(\boldsymbol{\theta}, Z_k) - \boldsymbol{g}(\boldsymbol{\theta}', Z_k)\| \leq 2\phi_\infty^2 \|\boldsymbol{\theta} - \boldsymbol{\theta}'\|, \text{ and}$$
$$\|\bar{\boldsymbol{g}}(\boldsymbol{\theta}) - \bar{\boldsymbol{g}}(\boldsymbol{\theta}')\| \leq 2\phi_\infty^2 \|\boldsymbol{\theta} - \boldsymbol{\theta}'\|.$$

**Lemma E.6.** *For any $0 \leq k' \leq k \leq T-1$, we have*

$$\mathbb{E}[\ell_{k'}\ell_k] \leq r_\infty^2\phi_\infty^2 + 2r_\infty\phi_\infty^3\mathbb{E}[\|\boldsymbol{\theta}_k\|] + 2r_\infty\phi_\infty^3\mathbb{E}[\|\boldsymbol{\theta}_{k'}\|] + 2\phi_\infty^4\mathbb{E}\left[\|\boldsymbol{\theta}_k\|^2\right] + 2\phi_\infty^4\mathbb{E}\left[\|\boldsymbol{\theta}_{k'}\|^2\right],$$

$$\mathbb{E}\left[\phi_\infty^2 d_{k'}\ell_k\right] \leq \frac{\phi_\infty^4 d_0^2 + r_\infty^2\phi_\infty^2}{2} + 2r_\infty\phi_\infty^3\mathbb{E}[\|\boldsymbol{\theta}_k\|] + 2\phi_\infty^4\mathbb{E}\left[\|\boldsymbol{\theta}_k\|^2\right] + \phi_\infty^4 d_0\mathbb{E}[\|\boldsymbol{\theta}_{k'}\|]$$
$$+ \frac{\phi_\infty^4}{2}\mathbb{E}\left[\|\boldsymbol{\theta}_{k'}\|^2\right].$$

*Proof.* From the AM-GM inequality, it suffices to upper bound $\mathbb{E}\left[\ell_k^2\right]$ and $\mathbb{E}\left[d_{k'}^2\right]$. We have

$$\mathbb{E}\left[\ell_k^2\right] = \mathbb{E}\left[(r_\infty\phi_\infty + 2\phi_\infty^2\|\boldsymbol{\theta}_k\|)^2\right] \leq r_\infty^2\phi_\infty^2 + 4r_\infty\phi_\infty^3\mathbb{E}[\|\boldsymbol{\theta}_k\|] + 4\phi_\infty^4\mathbb{E}\left[\|\boldsymbol{\theta}_k\|^2\right],$$
$$\mathbb{E}\left[d_{k'}^2\right] \leq \mathbb{E}\left[(\|\boldsymbol{\theta}_{k'}\| + d_0)^2\right] = d_0^2 + 2d_0\mathbb{E}[\|\boldsymbol{\theta}_{k'}\|] + \mathbb{E}\left[\|\boldsymbol{\theta}_{k'}\|^2\right].$$

Thus, we have

$$2\mathbb{E}[\ell_{k'}\ell_k] \leq \mathbb{E}\left[\ell_k^2\right] + \mathbb{E}\left[\ell_{k'}^2\right]$$
$$\leq 2r_\infty^2\phi_\infty^2 + 4r_\infty\phi_\infty^3\mathbb{E}[\|\boldsymbol{\theta}_k\|] + 4r_\infty\phi_\infty^3\mathbb{E}[\|\boldsymbol{\theta}_{k'}\|] + 4\phi_\infty^4\mathbb{E}\left[\|\boldsymbol{\theta}_k\|^2\right] + 4\phi_\infty^4\mathbb{E}\left[\|\boldsymbol{\theta}_{k'}\|^2\right],$$
$$2\mathbb{E}\left[\phi_\infty^2 d_{k'}\ell_k\right] \leq \mathbb{E}\left[\ell_k^2\right] + \mathbb{E}\left[\phi_\infty^4 d_{k'}^2\right]$$
$$\leq r_\infty^2\phi_\infty^2 + 4r_\infty\phi_\infty^3\mathbb{E}[\|\boldsymbol{\theta}_k\|] + 4\phi_\infty^4\mathbb{E}\left[\|\boldsymbol{\theta}_k\|^2\right] + \phi_\infty^4 d_0^2 + 2\phi_\infty^4 d_0\mathbb{E}[\|\boldsymbol{\theta}_{k'}\|]$$
$$+ \phi_\infty^4\mathbb{E}\left[\|\boldsymbol{\theta}_{k'}\|^2\right]. \qquad \square$$

## F   Bias and Gradient Norm

Recall that we decompose the updates as follows:

$$\boldsymbol{\theta}_t = \boldsymbol{\theta}_{t-1} + \eta_{t-1}\left(\underbrace{\boldsymbol{g}_{t-1} - \mathbb{E}[\boldsymbol{g}_{t-1} \mid \mathcal{F}_{t-2}]}_{\boldsymbol{\xi}_{t-1}} + \underbrace{\mathbb{E}[\boldsymbol{g}_{t-1} \mid \mathcal{F}_{t-2}] - \bar{\boldsymbol{g}}(\boldsymbol{\theta}_{t-1})}_{\boldsymbol{b}_{t-1}} + \bar{\boldsymbol{g}}(\boldsymbol{\theta}_{t-1})\right).$$

Then, we aim to bound terms on the right-hand side of the following inequality:

$$\mathbb{E}\left[\|\boldsymbol{\theta}_t - \boldsymbol{\theta}^*\|^2\right] \leq 2\mathbb{E}\left[\sum_{k=0}^{t-1} \eta_k\langle\boldsymbol{b}_k, \boldsymbol{\theta}_k - \boldsymbol{\theta}^*\rangle\right] + 3\mathbb{E}\left[\sum_{k=0}^{t-1} \eta_k^2 \|\boldsymbol{\xi}_k\|^2 + \sum_{k=0}^{t-1} \eta_k^2 \|\boldsymbol{b}_k\|^2 + \sum_{k=0}^{t-1} \eta_k^2 \|\bar{\boldsymbol{g}}(\boldsymbol{\theta}_k)\|^2\right]$$
$$+ \|\boldsymbol{\theta}^*\|^2.$$

We will bound each term separately in the following lemmas.

**Lemma F.1.** *For any $1 \leq t \leq T$, define $u_t := \left\lceil \frac{\log(2\sqrt{t})}{\log(1/\alpha)} \right\rceil$, we have*

$$\mathbb{E}\left[\sum_{k=0}^{t-1} \eta_k\langle\boldsymbol{b}_k, \boldsymbol{\theta}_k - \boldsymbol{\theta}^*\rangle\right] \leq 8\frac{d_0\ell_0\sqrt{u_t+1}}{c\phi_\infty^2 \log T} + 4\mathbb{E}\left[\sum_{k=u_t+1}^{t-1} \frac{d_{k-u_t}\ell_{k-u_t}}{c\phi_\infty^2 \log T \log(k+3)\sqrt{k+1}\sqrt{t}}\right]$$
$$+ \frac{2}{c^2\phi_\infty^4 \log^2 T}\mathbb{E}\left[\sum_{k=0}^{u_t} \frac{\ell_k + 2\phi_\infty^2 d_0}{\sqrt{k+1}\log(k+3)} \sum_{i=1}^{k} \frac{\ell_{i-1}}{\log(i+2)\sqrt{i}}\right]$$
$$+ \frac{2}{c^2\phi_\infty^4 \log^2 T}\mathbb{E}\left[\sum_{k=u_t+1}^{t-1} \frac{\ell_k + 2\phi_\infty^2 d_{k-u_t}}{\log(k+3)\log(k-u_t+3)\sqrt{k+1}} \sum_{i=k-u_t+1}^{k} \frac{\ell_{i-1}}{\sqrt{i}}\right].$$

*In particular, if* $\log T \geq \frac{2}{\log^3(1/\alpha)}$, *then by Lemma E.4,* $\sqrt{u_t + 1}/\log T \leq 1$. *Simplifying terms using Lemma E.6, we have* $\mathbb{E}\left[\sum_{k=0}^{t-1} \eta_k \langle \boldsymbol{b}_k, \boldsymbol{\theta}_k - \boldsymbol{\theta}^* \rangle\right] = \mathcal{O}(\max_{i \leq t-1} \mathbb{E}\left[\|\boldsymbol{\theta}_i\|^2\right] + \|\boldsymbol{\theta}^*\|^2)$.

*Proof.* Let $h_k := \min\{k, u_t\}$. We decompose the bias term as suggested in the proof sketch:

$$
\mathbb{E}\left[\sum_{k=0}^{t-1} \eta_k \langle \boldsymbol{b}_k, \boldsymbol{\theta}_k - \boldsymbol{\theta}^* \rangle\right] = \mathbb{E}\left[\sum_{k=0}^{t-1} \eta_k \langle \mathbb{E}[\boldsymbol{g}(\boldsymbol{\theta}_{k-h_k}, Z_k) \mid \mathcal{F}_{k-1}] - \bar{\boldsymbol{g}}(\boldsymbol{\theta}_{k-h_k}), \boldsymbol{\theta}_{k-h_k} - \boldsymbol{\theta}^* \rangle\right]
$$
$$
+ \mathbb{E}\left[\sum_{k=0}^{t-1} \eta_k \langle \mathbb{E}[\boldsymbol{g}(\boldsymbol{\theta}_k, Z_k) \mid \mathcal{F}_{k-1}] - \bar{\boldsymbol{g}}(\boldsymbol{\theta}_k), \boldsymbol{\theta}_k - \boldsymbol{\theta}^* \rangle\right]
$$
$$
- \mathbb{E}\left[\sum_{k=0}^{t-1} \eta_k \langle \mathbb{E}[\boldsymbol{g}(\boldsymbol{\theta}_{k-h_k}, Z_k) \mid \mathcal{F}_{k-1}] - \bar{\boldsymbol{g}}(\boldsymbol{\theta}_{k-h_k}), \boldsymbol{\theta}_{k-h_k} - \boldsymbol{\theta}^* \rangle\right] .
$$

For the $\mathbb{E}\left[\sum_{k=0}^{t-1} \eta_k \langle \mathbb{E}[\boldsymbol{g}(\boldsymbol{\theta}_{k-h_k}, Z_k) \mid \mathcal{F}_{k-1}] - \bar{\boldsymbol{g}}(\boldsymbol{\theta}_{k-h_k}), \boldsymbol{\theta}_{k-h_k} - \boldsymbol{\theta}^* \rangle\right]$ term above, we have

$$
\mathbb{E}\left[\sum_{k=0}^{t-1} \eta_k \langle \mathbb{E}[\boldsymbol{g}(\boldsymbol{\theta}_{k-h_k}, Z_k) \mid \mathcal{F}_{k-1}] - \bar{\boldsymbol{g}}(\boldsymbol{\theta}_{k-h_k}), \boldsymbol{\theta}_{k-h_k} - \boldsymbol{\theta}^* \rangle\right]
$$
$$
= \mathbb{E}\left[\sum_{k=0}^{t-1} \eta_k \langle \mathbb{E}[\boldsymbol{g}(\boldsymbol{\theta}_{k-h_k}, Z_k) \mid \mathcal{F}_{k-1}] - \bar{\boldsymbol{g}}(\boldsymbol{\theta}_{k-h_k}), \boldsymbol{\theta}_{k-h_k} - \boldsymbol{\theta}^* \rangle \,\Big|\, \mathcal{F}_{k-1-h_k}\right]
$$
$$
= \sum_{k=0}^{t-1} \mathbb{E}[\eta_k \langle \mathbb{E}[\boldsymbol{g}(\boldsymbol{\theta}_{k-h_k}, Z_k) - \bar{\boldsymbol{g}}(\boldsymbol{\theta}_{k-h_k}) \mid \mathcal{F}_{k-1-h_k}], \boldsymbol{\theta}_{k-h_k} - \boldsymbol{\theta}^* \rangle]
$$
$$
\leq \sum_{k=0}^{t-1} \mathbb{E}\left[\frac{1}{c\phi_\infty^2 \log T \log(k+3)\sqrt{k+1}} \|\boldsymbol{\theta}_{k-h_k} - \boldsymbol{\theta}^*\| \, \|\mathbb{E}[\boldsymbol{g}(\boldsymbol{\theta}_{k-h_k}, Z_k) - \bar{\boldsymbol{g}}(\boldsymbol{\theta}_{k-h_k}) \mid \mathcal{F}_{k-1-h_k}]\|\right]
$$
$$
\leq \sum_{k=0}^{t-1} \mathbb{E}\left[\frac{2d_{k-h_k} \sup_o \|\boldsymbol{g}(\boldsymbol{\theta}_{k-h_k}, o)\| \, C\alpha^{h_k}}{c\phi_\infty^2 \log T \log(k+3)\sqrt{k+1}}\right] \leq \sum_{k=0}^{t-1} \mathbb{E}\left[\frac{4d_{k-h_k} \ell_{k-h_k} \alpha^{h_k}}{c\phi_\infty^2 \log T \log(k+3)\sqrt{k+1}}\right]
$$
$$
\leq 8\frac{d_0 \ell_0 \sqrt{u_t + 1}}{c\phi_\infty^2 \log T} + 4\mathbb{E}\left[\sum_{k=u_t+1}^{t-1} \frac{d_{k-u_t} \ell_{k-u_t}}{c\phi_\infty^2 \log T \log(k+3)\sqrt{k+1}\sqrt{t}}\right],
$$

where we use Cauchy-Schwarz inequality in the first inequality and Lemma 5.3 in the second inequality. And for $k \geq u_t$, we have $\alpha^{h_k} \leq \frac{1}{\sqrt{t}}$, we use it with $C \leq 2$ in the last inequality. For the remaining term, we have

$$
\mathbb{E}\left[\sum_{k=0}^{t-1} \eta_k \langle \mathbb{E}[\boldsymbol{g}(\boldsymbol{\theta}_k, Z_k) \mid \mathcal{F}_{k-1}] - \bar{\boldsymbol{g}}(\boldsymbol{\theta}_k), \boldsymbol{\theta}_k - \boldsymbol{\theta}^* \rangle \right] - \mathbb{E}\left[\sum_{k=0}^{t-1} \eta_k \langle \mathbb{E}[\boldsymbol{g}(\boldsymbol{\theta}_{k-h_k}, Z_k) \mid \mathcal{F}_{k-1}] - \bar{\boldsymbol{g}}(\boldsymbol{\theta}_{k-h_k}), \boldsymbol{\theta}_{k-h_k} - \boldsymbol{\theta}^* \rangle \right]
$$

$$
= \mathbb{E}\left[\sum_{k=0}^{t-1} \eta_k \left( \Xi(\boldsymbol{\theta}_k, Z_k) - \Xi(\boldsymbol{\theta}_{k-h_k}, Z_k) \right) \right]
$$

$$
\leq \sum_{k=0}^{t-1} \mathbb{E}\left[ \eta_k (2\ell_k + 4\phi_\infty^2 d_{k-h_k}) \|\boldsymbol{\theta}_k - \boldsymbol{\theta}_{k-h_k}\| \right]
$$

$$
\leq \sum_{k=0}^{t-1} \mathbb{E}\left[ \eta_k (2\ell_k + 4\phi_\infty^2 d_{k-h_k}) \sum_{i=k-h_k}^{k-1} \eta_i \|\boldsymbol{g}_i\| \right]
$$

$$
= \mathbb{E}\left[ \sum_{k=0}^{u_t} \eta_k (2\ell_k + 4\phi_\infty^2 d_{k-h_k}) \sum_{i=k-h_k}^{k-1} \eta_i \|\boldsymbol{g}_i\| \right] + \mathbb{E}\left[ \sum_{k=u_t+1}^{t-1} \eta_k (2\ell_k + 4\phi_\infty^2 d_{k-h_k}) \sum_{i=k-h_k}^{k-1} \eta_i \|\boldsymbol{g}_i\| \right]
$$

$$
= \mathbb{E}\left[ \sum_{k=0}^{u_t} \frac{1}{c\phi_\infty^2 \log T \log(k+3)\sqrt{k+1}} (2\ell_k + 4\phi_\infty^2 d_0) \sum_{i=0}^{k-1} \frac{1}{c\phi_\infty^2 \log T \log(i+3)\sqrt{i+1}} \|\boldsymbol{g}_i\| \right]
$$

$$
+ \mathbb{E}\left[ \sum_{k=u_t+1}^{t-1} \frac{1}{c\phi_\infty^2 \log T \log(k+3)\sqrt{k+1}} (2\ell_k + 4\phi_\infty^2 d_{k-u_t}) \sum_{i=k-u_t}^{k-1} \frac{1}{c\phi_\infty^2 \log T \log(i+3)\sqrt{i+1}} \|\boldsymbol{g}_i\| \right]
$$

$$
\leq \frac{2}{c^2 \phi_\infty^4 \log^2 T} \mathbb{E}\left[ \sum_{k=0}^{u_t} \frac{\ell_k + 2\phi_\infty^2 d_0}{\sqrt{k+1}\log(k+3)} \sum_{i=1}^{k} \frac{\ell_{i-1}}{\log(i+2)\sqrt{i}} \right]
$$

$$
+ \frac{2}{c^2 \phi_\infty^4 \log^2 T} \mathbb{E}\left[ \sum_{k=u_t+1}^{t-1} \frac{\ell_k + 2\phi_\infty^2 d_{k-u_t}}{\log(k+3)\log(k-u_t+3)\sqrt{k+1}} \sum_{i=k-u_t+1}^{k} \frac{\ell_{i-1}}{\sqrt{i}} \right],
$$

where the first inequality comes from the Lemma 5.4. Putting it all together, we obtain the desired result. $\square$

**Lemma F.2.** *Define* $B_{t-1} = \frac{1}{c^2 \phi_\infty^4 \log^2 T} \sum_{k=0}^{t-1} \frac{r_\infty^2 \phi_\infty^2 + 4r_\infty \phi_\infty^3 \mathbb{E}[\|\boldsymbol{\theta}_k\|] + 4\phi_\infty^4 \mathbb{E}[\|\boldsymbol{\theta}_k\|^2]}{\log^2(k+3)(k+1)}$. *Then, for any* $t \leq T$, *we have*

$$
\mathbb{E}\left[ \sum_{k=0}^{t-1} \eta_k^2 \|\boldsymbol{\xi}_k\|^2 \right] \leq 4B_{t-1}, \quad \mathbb{E}\left[ \sum_{k=0}^{t-1} \eta_k^2 \|\boldsymbol{b}_k\|^2 \right] \leq 4B_{t-1}, \quad \mathbb{E}\left[ \sum_{k=0}^{t-1} \eta_k^2 \|\bar{\boldsymbol{g}}(\boldsymbol{\theta}_k)\|^2 \right] \leq B_{t-1} .
$$

*Proof.* By definition, we have

$$
\mathbb{E}\left[ \sum_{k=0}^{t-1} \eta_k^2 \|\boldsymbol{\xi}_k\|^2 \right] \leq \frac{4}{c^2 \phi_\infty^4 \log^2 T} \mathbb{E}\left[ \sum_{k=0}^{t-1} \frac{\ell_k^2}{\log^2(k+3)(k+1)} \right]
$$

$$
\leq \frac{4}{c^2 \phi_\infty^4 \log^2 T} \sum_{k=0}^{t-1} \frac{r_\infty^2 \phi_\infty^2 + 4r_\infty \phi_\infty^3 \mathbb{E}[\|\boldsymbol{\theta}_k\|] + 4\phi_\infty^4 \mathbb{E}\left[\|\boldsymbol{\theta}_k\|^2\right]}{\log^2(k+3)(k+1)} .
$$

The proofs for the bounds involving $\boldsymbol{b}_k$ and $\bar{\boldsymbol{g}}(\boldsymbol{\theta}_k)$ are similar. $\square$

## G   Bounded Iterates

We can now give the formal statement and prove Theorem 4.2:

**Theorem G.1.** *Let the horizon $T$ satisfy $\log T \geq \frac{2}{\log^3(1/\alpha)}$. Consider the unprojected TD learning algorithm initialized at $\boldsymbol{\theta}_0 = \mathbf{0}$ with the stepsize schedule*

$$\eta_t = \frac{1}{c\phi_\infty^2 \log T \log(t+3)\sqrt{t+1}},$$

*where $c > 15 + 18\sqrt{2}$. Then, for any $0 \leq t \leq T$, the iterates satisfy*

$$\mathbb{E}\left[\|\boldsymbol{\theta}_t\|^2\right] \leq \rho_c^2 \max\left\{\frac{r_\infty^2}{\phi_\infty^2}, \|\boldsymbol{\theta}^*\|^2\right\},$$

*where $\rho_c$ is a constant defined as:*

$$\rho_c = \frac{2c^2 + 36c + 434}{2(c^2 - 30c - 423)} + \sqrt{\frac{4c^4 + 256c^3 + 128c^2 - 29808c - 4532}{4(c^2 - 30c - 423)^2}}.$$

*Furthermore, $\rho_c$ is strictly decreasing in $c$, with $\lim_{c\to\infty} \rho_c = 2$ and $\lim_{c\to(15+18\sqrt{2})^+} \rho_c = \infty$.*

*Proof.* We recall the following estimates:

$$\begin{aligned}
\mathbb{E}\left[d_t^2 - d_0^2\right] &\leq 2\mathbb{E}\left[\sum_{k=0}^{t-1} \eta_k \langle \boldsymbol{b}_k, \boldsymbol{\theta}_k - \boldsymbol{\theta}^*\rangle\right] + 3\mathbb{E}\left[\sum_{k=0}^{t-1} \eta_k^2 \|\boldsymbol{\xi}_k\|^2 + \sum_{k=0}^{t-1} \eta_k^2 \|\boldsymbol{b}_k\|^2 + \sum_{k=0}^{t-1} \eta_k^2 \|\bar{\boldsymbol{g}}(\boldsymbol{\theta}_k)\|^2\right] \\
&\leq 16\frac{d_0\ell_0\sqrt{u_t+1}}{c\phi_\infty^2 \log T} + 8\mathbb{E}\left[\sum_{k=u_t+1}^{t-1} \frac{d_{k-u_t}\ell_{k-u_t}}{c\phi_\infty^2 \log T \log(k+3)\sqrt{k+1}\sqrt{t}}\right] \\
&\quad + \frac{4}{c^2\phi_\infty^4 \log^2 T}\mathbb{E}\left[\sum_{k=0}^{u_t} \frac{\ell_k + 2\phi_\infty^2 d_0}{\sqrt{k+1}\log(k+3)} \sum_{i=1}^{k} \frac{\ell_{i-1}}{\log(i+2)\sqrt{i}}\right] \\
&\quad + \frac{4}{c^2\phi_\infty^4 \log^2 T}\mathbb{E}\left[\sum_{k=u_t+1}^{t-1} \frac{\ell_k + 2\phi_\infty^2 d_{k-u_t}}{\log(k+3)\log(k-u_t+3)\sqrt{k+1}} \sum_{i=k-u_t+1}^{k} \frac{\ell_{i-1}}{\sqrt{i}}\right] \\
&\quad + \frac{27}{c^2\phi_\infty^4 \log^2 T}\sum_{k=0}^{t-1} \frac{r_\infty^2\phi_\infty^2 + 4r_\infty\phi_\infty^3\mathbb{E}[\|\boldsymbol{\theta}_k\|] + 4\phi_\infty^4\mathbb{E}\left[\|\boldsymbol{\theta}_k\|^2\right]}{\log^2(k+3)(k+1)}.
\end{aligned}$$

We use mathematical induction to show that $\mathbb{E}\left[\|\boldsymbol{\theta}_t\|^2\right] \leq \rho_c^2 \max\left\{\frac{r_\infty^2}{\phi_\infty^2}, d_0^2\right\}$ for all $t \leq T$. For base case $t = 0$:

$$\mathbb{E}\left[\|\boldsymbol{\theta}_0\|^2\right] = 0 \leq \rho_c^2 \max\left\{\frac{r_\infty^2}{\phi_\infty^2}, d_0^2\right\}.$$

Now, consider the induction hypothesis for $t \leq T$:

$$\max_{0 \leq i \leq t-1} \mathbb{E}\left[\|\boldsymbol{\theta}_i\|^2\right] \leq \rho_c^2 \max\left\{\frac{r_\infty^2}{\phi_\infty^2}, d_0^2\right\}.$$

Our goal is to show that $\mathbb{E}\left[\|\boldsymbol{\theta}_t\|^2\right] \leq \rho_c^2 \max\{\frac{r_\infty^2}{\phi_\infty^2}, d_0^2\}$.

Consider the case that $\frac{r_\infty}{\phi_\infty} \leq d_0$, and use Lemma E.6 with the induction hypothesis, to have

$$\mathbb{E}[\ell_{k'}\ell_k] \leq \beta_1 := r_\infty^2\phi_\infty^2 + 4r_\infty\phi_\infty^3\rho_c d_0 + 4\phi_\infty^4\rho_c^2 d_0^2,$$

$$\mathbb{E}\left[\phi_\infty^2 d_{k'}\ell_k\right] \leq \beta_2 := \frac{\phi_\infty^4 d_0^2 + r_\infty^2\phi_\infty^2}{2} + 2r_\infty\phi_\infty^3\rho_c d_0 + \phi_\infty^4\rho_c d_0^2 + 2.5\phi_\infty^4\rho_c^2 d_0^2.$$

Then, it follows that

$$\mathbb{E}\big[d_t^2\big] \le d_0^2 + \frac{16}{c\phi_\infty^2}\frac{d_0 r_\infty \phi_\infty}{\log T}\sqrt{u_t+1} + \frac{12}{c\phi_\infty^4 \log T}\beta_2$$
$$+ \frac{3(\sqrt{u_t+1})^3}{c^2\phi_\infty^4 \log^2 T}(\beta_1 + 2\beta_2)$$
$$+ \frac{8(u_t+1)}{c^2\phi_\infty^4 \log^2 T}(\beta_1 + 2\beta_2)$$
$$+ \frac{81}{c^2\phi_\infty^4 \log^2 T}\beta_1,$$

where we used Lemma E.1, Lemma E.2 and Lemma E.3 in the inequality. Recall that $u_t = \left\lceil \frac{\log(2\sqrt{t})}{\log(1/\alpha)} \right\rceil$ and $\log T \ge \frac{2}{\log^3(1/\alpha)}$. By Lemma E.4, we have $\sqrt{u_t+1}/\log T \le 1$, $\frac{u_t+1}{\log^2 T} \le 1$ and $\frac{(u_t+1)^{3/2}}{\log^2 T} \le 1$. Thus,

$$\mathbb{E}\big[d_t^2\big] \le d_0^2 + \frac{16d_0^2}{c} + \frac{12}{c}(1 + 3\rho_c + 2.5\rho_c^2)d_0^2$$
$$+ \frac{3}{c^2}(3 + 10\rho_c + 9\rho_c^2)d_0^2$$
$$+ \frac{8}{c^2}(3 + 10\rho_c + 9\rho_c^2)d_0^2$$
$$+ \frac{81}{c^2}(1 + 4\rho_c + 4\rho_c^2)d_0^2$$
$$\le d_0^2\left(1 + \frac{28 + 36\rho_c + 30\rho_c^2}{c} + \frac{114 + 434\rho_c + 423\rho_c^2}{c^2}\right).$$

That is,

$$\mathbb{E}\big[\|\boldsymbol{\theta}_t\|^2\big] \le \left(d_0 + \sqrt{\mathbb{E}[d_t^2]}\right)^2$$
$$\le \left(d_0 + \sqrt{1 + \frac{28 + 36\rho_c + 30\rho_c^2}{c} + \frac{114 + 434\rho_c + 423\rho_c^2}{c^2}}\, d_0\right)^2.$$

To fulfill the induction, we require

$$1 + \sqrt{1 + \frac{28 + 36\rho_c + 30\rho_c^2}{c} + \frac{114 + 434\rho_c + 423\rho_c^2}{c^2}} \le \rho_c .$$

Solving the inequality, we have the following condition:

$$c > \frac{30 + 36\sqrt{2}}{2}, \quad \rho_c = \frac{2c^2 + 36c + 434}{2(c^2 - 30c - 423)} + \sqrt{\frac{4c^4 + 256c^3 + 128c^2 - 29808c - 4532}{4(c^2 - 30c - 423)^2}} .$$

Note that when $c \to 15 + 18\sqrt{2}$, $\rho_c \to \infty$. And when $c \to \infty$, $\rho_c \to 2$. The case $\frac{r_\infty}{\phi_\infty} > d_0$ can be proved similarly. This completes the induction and the proof. $\qquad\square$

**Proposition G.2.** *Let*
$$c_0 := 15 + 18\sqrt{2},$$

*and for all real $c$ with $c > c_0$ define*

$$\rho_c = \frac{2c^2 + 36c + 434}{2(c^2 - 30c - 423)} + \sqrt{\frac{4c^4 + 256c^3 + 128c^2 - 29808c - 4532}{4(c^2 - 30c - 423)^2}} .$$

*Then $\rho_c$ is strictly decreasing on $(c_0, \infty)$. In particular,*

$$\lim_{c \to c_0^+} \rho_c = +\infty, \qquad \lim_{c \to \infty} \rho_c = 2 .$$

*Proof.* Write $\rho_c = \frac{2c^2 + 36c + 434 + \sqrt{D(c)}}{2Q(c)}$ with $Q(c) = c^2 - 30c - 423$ and $D(c) = (2c^2 + 36c + 434)^2 + 4Q(c)(28c + 114)$. For $c > c_0$, we have $Q(c) > 0$ and $D(c) > 0$.

A direct differentiation gives

$$\rho_c'(c) = -\frac{H(c) + K(c)\sqrt{D(c)}}{2Q(c)^2\sqrt{D(c)}},$$

where $K(c) = 96c^2 + 2560c + 2208 > 0$ and $H(c) = 248c^4 + 7352c^3 + 117720c^2 + 492200c - 6168432$. On the domain $c > c_0$, we have $H(c) \geq 492200c - 6168432 > 0$, hence $H(c) + K(c)\sqrt{D(c)} > 0$. Since the denominator is positive, $\rho_c'(c) < 0$ for all $c > c_0$.

The limits $\rho_c \to \infty$ as $c \downarrow c_0$ and $\rho_c \to 2$ as $c \to \infty$ follow immediately from $Q(c) \downarrow 0$ and the leading terms of the numerator/denominator. $\qquad\square$

## H   Details on the Experiments

**Setting.** We use the Julia[6] programming language to experiment with a toy MDP, which has $n = 50$ states, $|\mathcal{A}| = 1$ action, discount factor $\gamma = 0.99$, and the transition matrix $\boldsymbol{P}$ is a directed ring: from state $i \in \{1, 2, \cdots, 50\}$,

$$P_{i,i} = 0.1, \qquad P_{i,i+1} = 0.6, \qquad P_{i,i-1} = 0.3, \qquad \text{(with indices modulo 50)}.$$

We start with $s_0 = 1$. The stationary distribution $\pi$ is uniform, hence $\boldsymbol{D} = \frac{1}{n}\mathbf{I}$. The rewards $r(s, s') \in [0, 1]$ and features $\boldsymbol{\Phi} \in \mathbb{R}^{n \times d}$ are drawn once using fixed seeds ($d = 5$). Optionally, columns of $\boldsymbol{\Phi}$ are rescaled to control the spectrum. The fixed-point $\boldsymbol{\theta}^*$ of the projected Bellman equation system is computed by solving

$$\boldsymbol{A}\,\boldsymbol{\theta}^* = \boldsymbol{c}, \qquad \boldsymbol{A} := \boldsymbol{\Phi}^\top \boldsymbol{D}\,(\mathbf{I} - \gamma\boldsymbol{P})\,\boldsymbol{\Phi}, \qquad \boldsymbol{c} := \boldsymbol{\Phi}^\top \boldsymbol{D}\,\boldsymbol{r},$$

where $\boldsymbol{r} \in \mathbb{R}^n$ is the expected reward vector.

**Algorithm and diagnostics.** We run TD(0) with

$$\boldsymbol{\theta}_0 = \mathbf{0}, \quad \boldsymbol{\theta}_{t+1} = \boldsymbol{\theta}_t + \eta_t\big(r(s_t, s_{t+1}) + \gamma\,\boldsymbol{\phi}(s_{t+1})^\top \boldsymbol{\theta}_t - \boldsymbol{\phi}(s_t)^\top \boldsymbol{\theta}_t\big)\boldsymbol{\phi}(s_t),$$

and evaluate the weighted average $\bar{\boldsymbol{\theta}}_T = \left(\sum_{k=0}^{T-1}\eta_k\right)^{-1}\sum_{k=0}^{T-1}\eta_k\boldsymbol{\theta}_k$. To probe the presence of bounded iterates as in Theorem G.1, we sweep a scalar knob $c > 0$ that sets the theory-inspired stepsizes

$$\eta_t := \frac{1}{c\phi_\infty^2\,\log T\,\log(t+3)\sqrt{t+1}},$$

where horizon $T = 10^7$. A smaller $c$ means a larger effective stepsize. For each $c$ we simulate 48 independent trajectories and aggregate three diagnostics:

1. Expected boundedness ratio: $\frac{\max_{i \leq T}\mathbb{E}\big[\|\boldsymbol{\theta}_i\|^2\big]}{\|\boldsymbol{\theta}^*\|^2}$, which is large if iterates blow up;

2. Divergence rate: A run is marked as diverged if $\max_{t \leq T}\|\boldsymbol{\theta}_t\|^2 > 10^{12}$.

3. Suboptimality gap: $\mathbb{E}\left[(1-\gamma)\|\boldsymbol{V}_{\bar{\boldsymbol{\theta}}_T} - \boldsymbol{V}_{\boldsymbol{\theta}^*}\|_{\boldsymbol{D}}^2 + \gamma\left\|\boldsymbol{V}_{\bar{\boldsymbol{\theta}}_T} - \boldsymbol{V}_{\boldsymbol{\theta}^*}\right\|_{\mathrm{Dir}}^2\right]$.

The results are in Figure 1.

**Experiments with constant stepsize.**   We also used the constant stepsize

$$\eta := \frac{1}{c\phi_\infty^2\,\log T\,\log(T+3)\sqrt{T+1}},$$

to validate the hypothesis that the behaviors we observe are not due to the time-varying stepsizes.

The results are in Figure 2.

---

[6]Code available at `https://anonymous.4open.science/r/TMLR_Robust_Unprojected_TD_Learning-AC63`

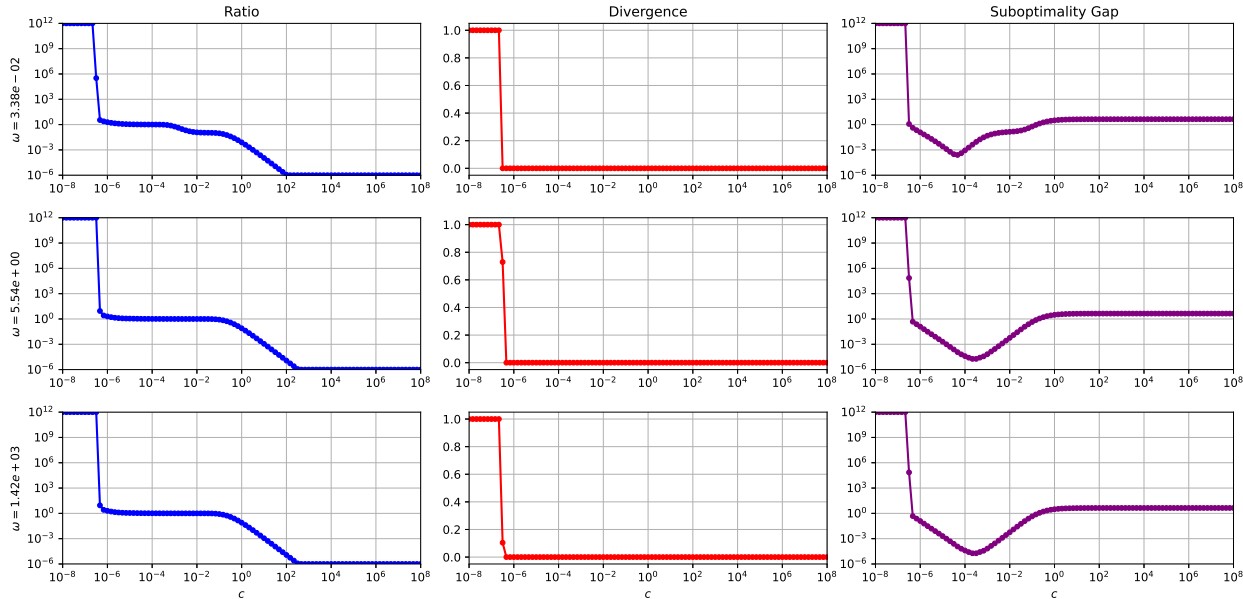

Figure 2: **Sweep over the stepsize scale $c$ (columns: boundedness ratio, divergence rate, suboptimality gap).** Rows correspond to different feature scalings, which change the spectrum of $\mathbf{\Phi}^\top \mathbf{D}\mathbf{\Phi}$ (the plot annotates the minimum eigenvalue and condition number for each row).

# I  Dependency on Mixing Time in Previous Algorithms

In this section, we clarify why Bhandari et al. (2018); Liu & Olshevsky (2021) exhibit an implicit dependence on the mixing time in their choice of $T$.

We describe it for Bhandari et al. (2018) only given that a similar reasoning holds for Liu & Olshevsky (2021). In Bhandari et al. (2018, Theorem 3), the convergence guarantee for projected TD(0) with linear function approximation takes the form

$$\mathbb{E}\left[\left\|\mathbf{V}_{\bar{\boldsymbol{\theta}}_T} - \mathbf{V}_{\boldsymbol{\theta}^*}\right\|_{\mathbf{D}}^2\right] \leq \frac{\left\|\boldsymbol{\theta}^* - \boldsymbol{\theta}_0\right\|^2 + G^2(9 + 12\tau(1/\sqrt{T}))}{2\sqrt{T}(1-\gamma)} \ .$$

Recall that $\tau(1/\sqrt{T}) \coloneqq \min\{t \in \mathbb{N} \mid C\alpha^t \leq \frac{1}{\sqrt{T}}\}$. To ensure that the bound is genuinely $\widetilde{\mathcal{O}}(1/\sqrt{T})$ without dependence on $C$ and $\alpha$, one can choose $T$ such that

$$\log T \geq \frac{1}{\log\left(\frac{1}{\alpha}\right)} \ , \quad \text{and } \sqrt{T} \geq C \ .$$

Thus we have $\tau(1/\sqrt{T}) = \left\lceil \frac{\log(C\sqrt{T})}{\log(1/\alpha)} \right\rceil \leq \log^2 T + 1$. This is analogous to our condition

$$\log T \geq \frac{C}{\log^3(1/\alpha)}$$

in Theorem G.1.

# J  Removing the Dependency on $T$ via the Doubling Trick

In this section, we explain how a standard doubling trick removes the requirement to know $T$ in our algorithm. This is a standard method, and we report it only for completeness.

Choose a stepsize based on an educated guess on the horizon $T'$ and run the algorithm for $T'$ iterations. After running $T'$ iterations, update the guess $T' \leftarrow 2T'$, restart the algorithm with $\boldsymbol{\theta}_0 = \mathbf{0}$, and set the stepsizes according to the new $T'$. This procedure repeats indefinitely. Now, at any moment in time, we have a number of models that were trained with an increasing number of samples, each one of them with the proposed learning rate, and each one of them starting from the zero vector. We might also have a model that is currently training, but it has not finished. We stress that these different models are independent of each other. So, it is enough to return the solution of the run that completed and used the most samples. If the samples used by this model are large enough, our convergence rate will apply to it.

What is the price that we pay in this way? Very minor: in the worst case, the solution we return was trained with at least half of the training samples. So, the rate will degrade only by a very small constant factor. Clearly, the factor '2' is also arbitrary and can be replaced by any factor greater than 1.

