# OpenReview forum: "A Robust $\widetilde{\mathcal{O}}(1/\sqrt{T})$ Rate for Unprojected TD Learning with Linear Function Approximation"
_TMLR — Under review for TMLR_

### Review · Reviewer_xRwF · 2026-05-21

**Summary Of Contributions:**

This paper provides a rigorous, finite-time convergence analysis of unprojected TD(0) with linear function approximation under Markovian noise. Crucially, the authors establish a robust $\tilde{\mathcal{O}}(1/\sqrt{T})$ convergence rate without relying on the artificial assumption of projecting the iterates onto a bounded set. By introducing a novel "self-bounding" property of the TD updates, the authors successfully resolve an open problem raised in prior literature (e.g., Bhandari et al., 2018) regarding the necessity of projection or additional regularity conditions in these settings.

**Audience:**

Yes

**Audience Explanation:**

Temporal Difference (TD) learning is a cornerstone algorithm in reinforcement learning, and understanding its finite-time convergence properties is an active area of research. Specifically, the paper resolves a known open problem raised in foundational prior work (Bhandari et al., 2018) by proving that an artificial iterate projection step is not required to achieve robust convergence under Markovian noise. Because projection is rarely used in practical implementations of TD learning, removing this assumption bridges an important gap between theoretical analysis and practical application.

**Broader Impact Concerns:**

As this is a purely theoretical paper establishing convergence properties of a foundational algorithm, it does not raise direct ethical concerns or immediate broader impact issues

**Claims And Evidence:**

Yes

**Claims Explanation:**

The primary claims made in this submission are theoretical, and they are supported by rigorous mathematical evidence. The authors provide formal theorems establishing the finite-time $\tilde{\mathcal{O}}(1/\sqrt{T})$ convergence rate for unprojected TD(0) under Markovian noise. These claims are backed by extensive proofs (detailed in the supplementary material) that clearly articulate the mechanics of the novel "self-bounding" property used to bypass the need for iterate projection.

Furthermore, the theoretical claims are accompanied by empirical evidence. While the experiments are highly restricted in scope—focusing on a synthetic two-state Markov chain rather than standard RL benchmarks—they are fit for purpose.

**Requested Changes:**

* In Theorem 4.2, the analysis requires the stepsize parameter to satisfy $c \ge 15+18\sqrt{2}$. While I appreciate the authors' transparency in stating that they "did not try to optimize the numerical value of the threshold on $c$," this mathematically safe constant results in an impractically small learning rate. It would significantly strengthen the paper if the authors could provide empirical guidance on practical choices for $c$, or discuss the gap between this theoretical requirement and what is actually needed for stability in practice.

* The proposed learning rate schedule explicitly depends on the total iteration budget $T$. While the authors address this by suggesting the standard "doubling trick" in the appendix, resetting model weights and restarting the algorithm periodically is inefficient and rarely done in practical online TD learning implementations. It might strengthen the submission if the authors can briefly discussed the potential for extending their analysis to "anytime" bounds (or adaptive step sizes without horizon knowledge).

---

> ### Author Response · Authors · 2026-06-01
>
> We thank the reviewer for the careful reading and for the positive assessment of our main contribution. We agree that the practical interpretation of the constant $c$ and the horizon-dependent stepsize deserves a clearer discussion.
>
> **On the threshold $c\geq 15+18\sqrt{2}$:**
>
> The threshold $ c>c_0=15+18\sqrt{2} $ should be viewed as a conservative sufficient condition for the proof, not as a practical tuning rule. This constant comes from the self-bounding induction used to prove bounded iterates for $t\leq T$. More specifically, it accumulates from geometrically mixing, the use of the integral test, repeated uses of the triangle inequality, Cauchy--Schwarz, and AM--GM type inequalities. Thus, $c$ is chosen large enough so that all these proof-level constants can be absorbed and the induction closes. In the revised version, we have added a paragraph explaining where $c$ is coming from and highlight it in red in the revised manuscript.
>
> Because our paper is primarily theoretical, our experiments are meant to provide qualitative evidence for the self-bounding phenomenon, not to prescribe a numerically optimal learning rate. We believe that providing a practical guidance on selecting $c$ is out of the scope of our theoretical contribution.
>
>
> **On fixed budget $T$:**
>
> We agree with the reviewer that the doubling trick is not how online TD is usually implemented. Indeed, the doubling trick in Appendix J is used only as a theoretical wrapper, not as a practical solution.
>
> The condition on $T$ that enters our theorem is related to the control of Markovian sampling bias. In the proof, we compare the current iterate with an older iterate $u_t$ steps in the past, where
> $$
>     u_t=\left\lceil\frac{\log(2\sqrt t)}{\log(1/\alpha)}\right\rceil,
> $$
> so that $ C\alpha^{u_t}\le 1/\sqrt t $. This introduces explicit mixing-related factors in the induction, namely
> $$
>     \frac{\sqrt{u_t+1}}{\log T},\qquad
>     \frac{u_t+1}{\log^2 T},\qquad
>     \frac{(u_t+1)^{3/2}}{\log^2 T}.
> $$
> The condition $ \log T\ge 2/\log^3(1/\alpha) $ is used to upper bound these quantities by universal constants. Without such a condition, one would need a stepsize explicitly depending on the mixing time. In the revised version, we have modified the paragraph "Dependency on $\alpha$ for $T$?" to make it clearer.

---

### Review · Reviewer_npW3 · 2026-05-22

**Summary Of Contributions:**

This theoretical paper found proof of a robust $O(1/\sqrt(T))$ rate for temporal difference (TD) learning with linear function approximation without projection. It resolves a longstanding and open problem from Bhandari et al (COLT18), while revealing a self-bounding property.
Instead of prior work which relied on contradiction based methods, this work found a direct proof which may generalize to other settings.

**Additional Comments:**

I am not an expert in RL so this is a little bit of an outside view.

**Audience:**

Yes

**Audience Explanation:**

The paper solves a concrete problem discussed in the literature with a clean result. It is well written and neatly illustrated.
The choice of the potential function in particular is interesting.
The fact that the problem is actively discussed servers as evidence of community interest.
Potentially the proof could also be if interest in related disciplines.

**Claims And Evidence:**

Yes

**Claims Explanation:**

this theoretical paper provides an original proof as evidence.
The proof appears novel and interesting, although a complete verification is beyond the scope of this review.
The experiments in table one add a neat empirical angle to the paper.

- Where is the threshold c Algorithm 1 and Theorem 4.2 coming from? Would it would be helpful to include an empirical estimate of the threshold, perhaps based on the existing experiments?

**Requested Changes:**

The supplementary material is quite extensive. As far as I can tell everything is covered.

---

> ### Author Response · Authors · 2026-06-01
>
> We thank the reviewer for the positive assessment of our contribution.
>
> **On the threshold $c\geq 15+18\sqrt{2}$:**
>
> The threshold $c>c_0=15+18\sqrt{2}$ should be viewed as a conservative sufficient condition for the proof, not as a practical tuning rule. This constant comes from the self-bounding induction used to prove bounded iterates for $t\leq T$. More specifically, it accumulates from geometrically mixing, the use of Integral test, repeated uses of the triangle inequality, Cauchy--Schwarz, and AM--GM type inequalities. Thus, $c$ is chosen large enough so that all these proof-level constants can be absorbed and the induction closes. In the revised version, we have added a paragraph explaining where $c$ is coming from.
>
> One can easily see the value of the threshold $c$ from our experiments.
> However, our paper is primarily theoretical, and our experiments are meant to provide qualitative evidence for the self-bounding phenomenon, not to prescribe a numerically optimal learning rate.

---

### Review · Reviewer_xvwu · 2026-05-24

**Summary Of Contributions:**

This paper studies finite-time convergence of unprojected TD(0) with linear function approximation under Markovian sampling. The main contribution is a robust convergence guarantee for the unprojected algorithm: the result avoids both the projection step used in earlier robust analyses and the dependence on the minimal curvature parameter that appears in fast-rate analyses. The proof centers on a self-bounding argument showing that, with a polylog-corrected stepsize, the TD iterates remain bounded in expectation. This boundedness result is then used to obtain a robust expected convergence rate for the Liu and Olshevsky potential. The paper frames this as resolving an open problem from Bhandari et al. (2018) in the robust-rate, unprojected setting.

Key strengths:

- The problem is well motivated. Projection is widely understood as an analytical tool rather than a practical part of TD learning, so removing it from robust finite-time analysis is meaningful.
- The claimed result fills a clear gap between projected robust TD analyses and unprojected curvature-dependent fast-rate analyses.
- The self-bounding proof strategy is interesting and may be useful beyond this specific TD(0) result.
- The paper provides a useful comparison between robust and fast rates, including examples where curvature-dependent bounds can be non-informative.
- The supplemental material contains the main missing technical details, including the bounded-iterate theorem, Markovian bias control, experiment setup, and doubling-trick argument.

Key weaknesses:

- The main paper relies heavily on the supplemental material. This is acceptable, but the most important result, the bounded-iterate theorem, is too central to be treated only as a deferred proof. The main text should include a sharper proof roadmap and more explicit assumptions/dependencies.
- The constants and dependencies hidden in `O~` notation remain hard to audit from the theorem statement alone, especially the dependence on the mixing parameters, discount factor, reward bound, feature bound, and the threshold parameter `c`.
- The algorithm and theorem presentation could be clearer. The stepsize formula is visually hard to parse in the main PDF, and the proof sketch contains a confusing phrase saying the stepsize parameter `c` should be "small enough" even though the theorem requires `c > c0`; the intended statement appears to be that the actual stepsize should be small enough, equivalently `c` large enough.
- The experiment is useful for intuition but limited. It supports the existence of a threshold phenomenon on a synthetic problem, but it is not evidence for the theorem beyond illustrating the behavior.

**Additional Comments:**

The paper is a good fit for TMLR as a focused theoretical contribution. My main recommendation is to improve presentation rather than substantially change the result. The supplemental material resolves my initial concern about missing proof details, but the main text should do more to make the central theorem and its dependencies transparent.

**Audience:**

Yes

**Audience Explanation:**

Yes. TMLR's audience includes researchers interested in the mathematical principles of learning systems, and this paper addresses a concrete theoretical gap in temporal-difference learning. A robust finite-time analysis of unprojected TD(0) under Markovian sampling is relevant to researchers working on reinforcement learning theory, stochastic approximation, and finite-time analysis of practical learning algorithms.

The contribution is specialized, but it is clearly within TMLR's scope. TMLR does not require broad significance if the work is correct, clearly supported, and of interest to a subset of the community. This paper meets that bar in topic and potential impact.

**Broader Impact Concerns:**

I do not see broader-impact concerns requiring a special broader impact statement. The submission is a theoretical analysis of TD learning with linear function approximation and does not introduce a new deployed system, dataset, data collection process, or application domain with immediate ethical risks. Standard RL deployment risks are not materially changed by this theoretical result.

**Claims And Evidence:**

Yes

**Claims Explanation:**

After reading the supplemental material, I believe the main claims are supported at the level expected for a theory paper, subject to the authors improving clarity. The supplement includes the key proof ingredients that are only sketched in the main text: the proof of Lemma 5.3, supporting technical lemmas, the detailed control of the Markovian bias and gradient-norm terms, the formal bounded-iterate proof in Theorem G.1, the experiment details, and the doubling-trick discussion. The proof structure is coherent: the authors control the bias induced by Markovian sampling using mixing, use this to bound the terms in the iterate recursion, and then close an induction proving bounded second moments of the iterates. This boundedness is then used in the convergence argument for the weighted average.

I do not see an obvious flaw in the high-level logic. The paper also accurately situates the result against prior robust projected analyses and fast curvature-dependent analyses. The theoretical evidence is therefore convincing enough for the claimed contribution, assuming the supplemental material is considered part of the submission.

That said, the paper should make the central assumptions and dependencies easier to verify. The result requires a horizon condition depending on the mixing parameter, and the final theorem hides several constants and dependencies. Since the contribution is a finite-time theorem, the authors should make these dependencies more explicit in the statement or immediately after it. I would also ask the authors to fix the minor but confusing wording around the stepsize threshold and make the Algorithm 1 stepsize formula unambiguous.

**Requested Changes:**

Critical to acceptance:

1. Ensure the complete supplemental material is included and clearly linked from the submission. The main theorem depends on Theorem G.1 and the Appendix F bias controls; these must remain available to reviewers and readers.
2. Clarify the stepsize definition in Algorithm 1. Please write it as an unambiguous displayed equation, e.g. `eta_t = 1 / (c phi_infty^2 log(T) log(t+3) sqrt(t+1))`, if that is the intended formula.
3. Fix the apparent inconsistency in Section 5 where the proof sketch says the induction works "if the stepsize parameter c is small enough," while Theorem 4.2 and Theorem G.1 require `c > c0`. This should be rewritten as a statement about the actual stepsize being small enough, or about `c` being sufficiently large.
4. State more explicitly what is hidden by the `O~` notation in Theorem 4.2(b), especially dependencies on the mixing constants, `gamma`, `r_infty`, `phi_infty`, and `c`.
5. Clarify the horizon/mixing-time condition in Theorem 4.2. The paper should explain in the theorem statement or immediately afterward exactly how large `T` must be relative to the mixing parameter and what happens outside that regime.

Would strengthen the work:

1. Add a concise proof roadmap before or after Theorem 4.2 that identifies the core novelty relative to Bhandari et al. (2018) and Liu and Olshevsky (2021).
2. Move a simplified version of Theorem G.1 or its induction invariant into the main text. The bounded-iterate result is central enough that readers should not need to go directly to the appendix to understand its shape.
3. Add a small table or paragraph comparing the final bound's dependencies with the most closely related projected robust bounds.
4. Improve notation consistency around `g_t`, `g_t(theta)`, `Z_t`, filtrations, and the shifted indices in the Markovian-bias argument.
5. Expand the discussion of the synthetic experiment limitations. The plots are helpful, but the paper should not overstate them as evidence beyond illustrating the threshold behavior.

---

> ### Author Response · Authors · 2026-06-01
>
> We thank the reviewer for the careful and constructive review. We agree with the main message of the reviewer that the presentation should be easier to audit. We have changed the paper following the suggested modifications and highlighted the changes in red.
>
> **Ensure the complete supplemental material is included and clearly linked:**
>
> We thank the reviewer for the reminder. In the revised version, we have added the complete PDF version including the appendices to ensure Theorem G.1 and the bias controls  in Appendix F are readily available to reviewers and readers.
>
>
> **Clarify the stepsize definition in Algorithm 1:**
> We thank the reviewer for pointing this out. In the revised version, we have revised Algorithm 1 to state the stepsize as the explicit displayed equation.
>
> **Fix $c$ being sufficiently small:**
> Thank you for catching this inconsistency. In the revised version, the statement in the proof sketch has been corrected from “$c$ is small enough” to “$c$ is large enough”.
>
> **State more explicitly what is hidden by the `O~` notation:**
>
> We agree with the reviewer that the statement in Theorem 4.2(b) should be more explicit. In the revised version, we have renamed Theorem 4.2(b) as Corollary 4.3 and stressed that  $\widetilde{\mathcal{O}}$ hides only logarithmic factors in $T$. It hides no dependence on the mixing constant $\alpha$ or the discount factor $\gamma$.
>
>
> **Clarify the horizon/mixing-time condition in Theorem 4.2:**
>
> We agree with the reviewer that the presentation of horizon/mixing-time condition in Theorem 4.2 should be more clear. In the revised version, we have modified the paragraph "Dependency on $\alpha$ for $T$?" to make it more clear.
>
> **Proof roadmap before or after Theorem 4.2 that identifies the core novelty relative to Bhandari et al. (2018) and Liu and Olshevsky (2021); Simplified version of Theorem G.1 or its induction invariant into the main text:**
>
> Section 5 already contains the proof roadmap, contrasting it with previous approaches, as well as a simplified argument of the core idea. In the revised version, we have also made the argument around controlling the bias more clear. Please let us know if something is unclear or it should be improved and we will be happy to amend it.
>
> **Small table or paragraph comparing the final bound's dependencies with the related projected robust bounds:**
>
> We already list all the dependencies in Table 1, would the Reviewer prefer something different? Note that we have chosen to list the dependencies rather than the rates because some parameters are used in the algorithms.
>
> **Improve notation consistency:**
> We thank the reviewer for pointing this out. In the revised version, we have defined the TD update map in Section 3 and improved the notation consistency around `g_t`, `g_t(theta)`, `Z_t`, filtrations, and the shifted indices in the Markovian-bias argument.
>
> **Expand the discussion of the limitations of the synthetic experiment:**
>
> We agree with the reviewer that the experiment should be viewed only as illustrating the behavior of Theorem 4.2. In the revised version, we have modified the paragraph "Is the threshold on the stepsizes real?".